METHODS AND RESOURCES

# A multichannel electrophysiological approach to noninvasively and precisely record human spinal cord activity

Birgit Nierula[1¤a], Tilman Stephani[2,3,4¤b], Emma Bailey[1,4], Merve Kaptan[1,4¤c], Lisa-Marie Geertje Pohle[1,4], Ulrike Horn[1], André Mouraux[5], Burkhard Maess[6], Arno Villringer[3], Gabriel Curio[7], Vadim V. Nikulin[2,3☯], Falk Eippert[1]☯*

1 Max Planck Research Group Pain Perception, Max Planck Institute for Human Cognitive and Brain Sciences, Leipzig, Germany, 2 Research Group Neural Interactions and Dynamics, Department of Neurology, Max Planck Institute for Human Cognitive and Brain Sciences, Leipzig, Germany, 3 Department of Neurology, Max Planck Institute for Human Cognitive and Brain Sciences, Leipzig, Germany, 4 International Max Planck Research School NeuroCom, Leipzig, Germany, 5 Institute of Neuroscience, Université Catholique de Louvain, Brussels, Belgium, 6 Methods and Development Group Brain Networks, Max Planck Institute for Human Cognitive and Brain Sciences, Leipzig, Germany, 7 Department of Neurology, Charité University Medicine, Berlin, Germany

☯ These authors contributed equally to this work.
¤a Current address: Fraunhofer Heinrich-Hertz-Institute, Department for Vision and Imaging Technology, Interactive and Cognitive Systems Group, Berlin, Germany
¤b Current address: Donders Institute for Brain, Cognition and Behaviour, Radboud University, Nijmegen, the Netherlands
¤c Current address: Division of Pain Medicine, Department of Anesthesiology, Perioperative and Pain Medicine, Stanford University School of Medicine, Palo Alto, California, United States of America
* eippert@cbs.mpg.de.

**Data Availability Statement:** All raw data have been uploaded in EEG-BIDS format to OpenNeuro (Study 1: https://openneuro.org/datasets/

## Abstract

The spinal cord is of fundamental importance for integrative processing in brain–body communication, yet routine noninvasive recordings in humans are hindered by vast methodological challenges. Here, we overcome these challenges by developing an easy-to-use electrophysiological approach based on high-density multichannel spinal recordings combined with multivariate spatial-filtering analyses. These advances enable a spatiotemporal characterization of spinal cord responses and demonstrate a sensitivity that permits assessing even single-trial responses. To furthermore enable the study of integrative processing along the neural processing hierarchy in somatosensation, we expand this approach by simultaneous peripheral, spinal, and cortical recordings and provide direct evidence that bottom-up integrative processing occurs already within the spinal cord and thus after the first synaptic relay in the central nervous system. Finally, we demonstrate the versatility of this approach by providing noninvasive recordings of nociceptive spinal cord responses during heat-pain stimulation. Beyond establishing a new window on human spinal cord function at millisecond timescale, this work provides the foundation to study brain–body communication in its entirety in health and disease.

ds004388; Study 2: https://openneuro.org/datasets/ds004389; Study 3: https://openneuro.org/datasets/ds005307). The data underlying Figs 2, 3, 4, 5, 6, 8 and S1 are available via Supplementary Data Files. All analysis code has been deposited on GitHub and is openly available (see https://github.com/eippertlab/spinal_sep1, https://doi.org/10.5281/zenodo.13383050; https://github.com/eippertlab/spinal_sep2, https://doi.org/10.5281/zenodo.13383046; https://github.com/eippertlab/spinal-lep1, https://doi.org/10.5281/zenodo.13383056).

**Funding:** FE received funding from the Max Planck Society and the European Research Council (under the European Union's Horizon 2020 research and innovation programme; grant agreement No 758974). The funders had no role in study design, data collection and analysis, decision to publish, or preparation of the manuscript.

**Competing interests:** The authors have declared that no competing interests exist.

**Abbreviations:** CCA, canonical correlation analysis; CHEP, contact heat-evoked potential; ECG, electrocardiography; EEG, electroencephalography; EMG, electromyography; ENG, electroneurography; ESG, electrospinography; fMRI, functional magnetic resonance imaging; ICA, independent component analysis; IR, interaction ratio; LEP, laser-evoked potential; LM, left mastoid; LME, linear-mixed-effects; ML, maximum likelihood; MSG, magnetospinography; NAP, nerve action potential; OBS, optimal basis set; OPM, optically pumped magnetometer; PCA, principal component analysis; PEP, pinprick-evoked potential; PSDC, postsynaptic dorsal column; RM, right mastoid; SEP, somatosensory evoked potential; SNR, signal-to-noise ratio.

## Introduction

The spinal cord is an important interface for reciprocal brain–body communication in sensory, motor, and autonomic domains [1]. Traditionally, it has been portrayed as a relay station, yet recent studies challenge this long-held view, for example, in the somatosensory domain, where a high degree of neuronal complexity and circuit organization has been delineated in animal models, suggestive of extensive integrative processing [2–4]. Such advances are important in order to arrive at a mechanistic understanding of spinal processing, especially considering the spinal cord's central role in numerous neurological disorders [5–7] as well as in treatment approaches for spinal cord injury [8,9] or biomarker development for analgesic drug discovery [10,11]. While there is a continuous development of sophisticated spinal recording technologies in experimental animals [12,13], such progress is missing in human neuroscience and knowledge on processing in the human spinal cord is consequently very limited, thus presenting a missing link in a comprehensive understanding of brain–body communication in health and disease.

Approaches such as reflex recordings [14,15] allow for useful assessments of the processes occurring within the human spinal cord, yet they only provide an indirect picture and more direct assessments via neuroimaging techniques are highly desirable. Several factors make the spinal cord a very challenging target for noninvasive neuroimaging, however: It has a small diameter, is located deep in the body in close proximity to inner organs such as the heart and lungs, and is protected by the vertebral column and muscle layers. Consequently, there is a lack of well-established and readily available approaches to interrogate human spinal cord function. For example, functional magnetic resonance imaging (fMRI) of the human spinal cord [16] comes with major technical challenges [17] and is fundamentally limited by its indirect link to neuronal activity via neurovascular coupling and ensuing low temporal resolution. Magnetospinography (MSG) on the other hand is a noninvasive method that directly measures the magnetic fields generated by neuronal populations in the spinal cord with high temporal precision [18], yet no commercially available systems have been developed [19]. Both approaches are therefore only pursued by a small number of research groups and additionally require major investments in large-scale equipment, preventing their widespread use in human neuroscience.

Here, we introduce a novel approach that overcomes these issues. It is based on an enhancement of methodology established several decades ago during the development of noninvasive electrospinography (ESG) [20–24]. These studies recorded somatosensory evoked potentials (SEPs) from the human spinal cord via surface electrodes placed on the skin over the vertebral column and reported SEPs with a postsynaptic origin in the dorsal horn of the spinal cord [25–30]. While useful in clinical settings [31,32], due to technical challenges, this line of research has, however, largely subsided in experimental neuroscience, with only a handful of studies recording such spinal SEPs noninvasively in healthy human volunteers in the last decade [33–38].

To improve upon these approaches and expand the insights ESG can offer, we leveraged the developments that have occurred in recording capabilities and processing techniques for neurophysiological data [39–41]: We developed a noninvasive approach that allows for recording spinal signals with high temporal precision (10 kHz) as well as extensive spatial coverage (multichannel montage of 39 surface electrodes placed over the neck and trunk in two dense electrode grids) and combined this with concurrent recordings of the input to (peripheral nerve action potentials (NAPs)) and output from the spinal cord (brainstem and cortical SEPs). Furthermore, we developed dedicated artifact-correction techniques to enhance the

spinal signal-to-noise ratio (SNR) and employed multivariate analysis approaches that allowed for increased robustness as well as extraction of spinal cord responses at the single-trial level.

This approach thus provides a direct and easily accessible electrophysiological window into a previously missing link of brain–body communication relevant for several domains in human neuroscience. Here, we chose the domain of somatosensation as test-bed and employed this approach in two complementary studies (Fig 1), in both of which we recorded signals from the cervical and lumbar spinal cord, in order to allow for the generalization of our findings across upper and lower limb representations. Most importantly, this approach allowed us to directly investigate whether integrative processes already occur at the level of the human spinal cord, i.e., at the first station of central nervous system processing. In a final proof-of-principle experiment, we furthermore assessed the possibility of using this noninvasive approach to detect nociceptive spinal cord responses in humans.

## Results

### Delineating somatosensory responses along the neural hierarchy (Experiment 1)

As a first objective, we aimed to replicate previously reported somatosensory responses along the neural hierarchy, with a special focus on the spinal cord and thus simultaneously recorded

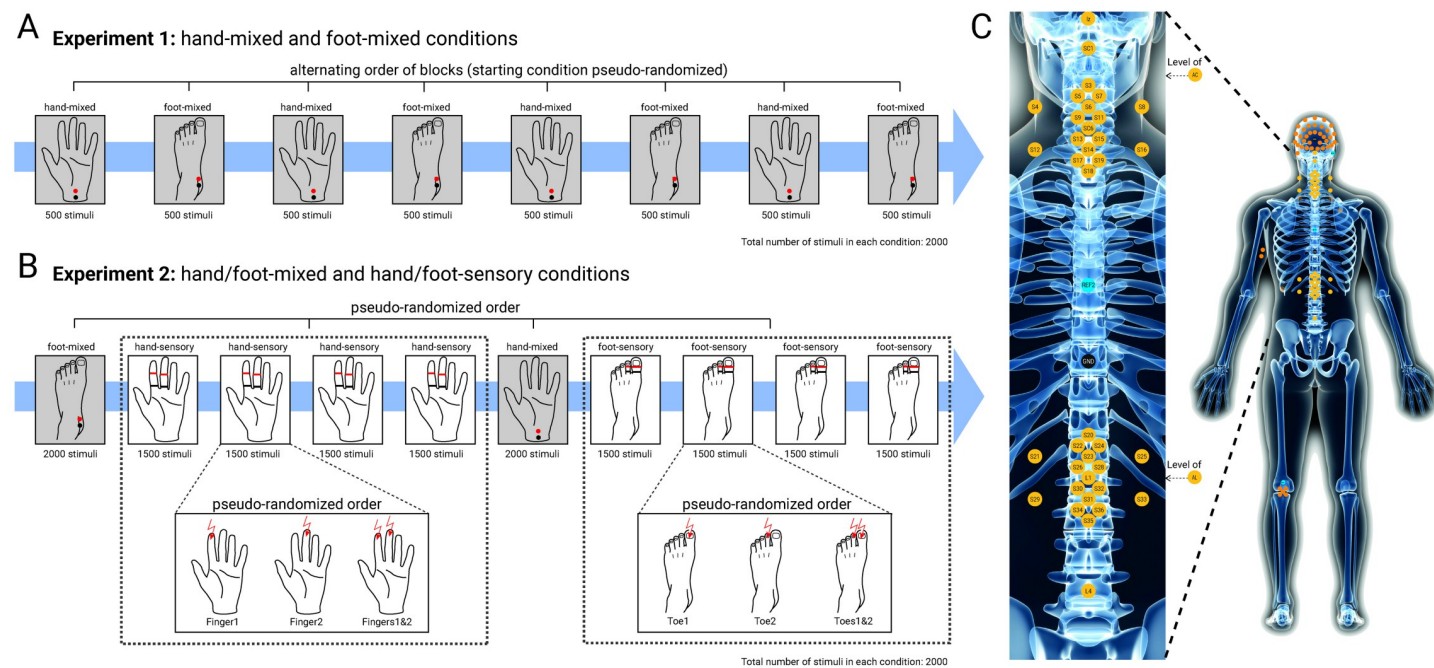

**Fig 1. Overview of experimental conditions and recording setup. (A)** In Experiment 1, electrical mixed nerve stimulation was applied to the left median nerve at the wrist (hand-mixed) and to the left tibial nerve at the ankle (foot-mixed). Four hand-mixed and four foot-mixed blocks were presented in alternating order. **(B)** In Experiment 2, electrical mixed nerve stimulation was applied to the same location as in Experiment 1, and electrical sensory nerve stimulation was applied to the left index and middle finger (hand-sensory) and to the first and second toe (foot-sensory). Sensory stimulation blocks were separated into four consecutive blocks of the same stimulation type (either hand-sensory or foot-sensory). **(C)** Across both experiments, responses were recorded at the level of the peripheral nerves, the spinal cord, and the brain. Peripheral NAPs were recorded from the ipsilateral axilla and Erb's point for median nerve stimulation and from the ipsilateral popliteal fossa (cluster of 5 electrodes) and the cauda equina for tibial nerve stimulation. Spinal cord SEPs were recorded with a montage of 37 dorsal and two ventral electrodes, which had a cervical and a lumbar focus: Around an anatomical target electrode (placed over the spinous process of either the sixth cervical vertebra or the first lumbar vertebra), 17 electrodes were placed in a grid with distances optimized for capturing the spatial distribution of the spinal signal. Additionally, the following electrodes were contained in the spinal montage: one over the inion, one over the first cervical vertebra, one over the spinous process of the fourth lumbar vertebra, and two ventral electrodes (AC located supraglottically and AL located supraumbilically). All electrodes of the spinal montage were referenced to an electrode placed over the spinous process of the sixth thoracic vertebra. Cortical SEPs were recorded with a 64-channel EEG setup in Experiment 1 (39 channels in Experiment 2).

peripheral NAPs as well as SEPs from the spinal cord, brainstem, and cortex to upper and lower limb stimulation. In the hand-mixed condition, we extracted the peripheral N6 (origin: median nerve), the peripheral N9 (origin: brachial plexus), the spinal N13 (origin: dorsal horn), the brainstem N14 (likely origin: cuneate nucleus), and the cortical N20 (origin: primary somatosensory cortex). In the foot-mixed condition, we extracted the peripheral N8 (origin: tibial nerve), the spinal N22 (origin: dorsal horn), the brainstem N30 (likely origin: gracile nucleus), and the cortical P40 (origin: primary somatosensory cortex).

Replication was successful at all recording sites, where we observed response amplitudes that were highly significant at the group level ($N = 36$) and exhibited consistently large effect sizes (Table 1); to furthermore ensure the robustness of these results, we replicated them in Experiment 2 (S1 Text and S1 Table). Grand-average time-courses at the group level are depicted in Fig 2 and delineate the temporal progression of the neurophysiological signal along the processing hierarchy, providing a robust and comprehensive view on somatosensory processing from periphery to cortex.

## Characterizing spinal SEPs in detail (Experiment 1)

Next, we aimed to provide a spatial, temporal, and spectral characterization of spinal responses. First, the grand-average time-course of the potentials obtained from single, anatomically defined target electrodes exhibited a triphasic shape with an initial positive deflection, a main negative deflection (at 13 ms and 24 ms, respectively), and a slowly decaying late positive deflection (red trace in Fig 3A and 3E). Second, our multichannel setup allowed for the first time to estimate the potentials' spatial topography (Fig 3B and 3F), which showed a radial dipole at peak latency, with a center over the spinal cord, close to the spinal segments targeted by the electrical stimulation at wrist and ankle. Importantly, the topographies show that N13 and N22 responses are consistently limited to the relevant electrode-grid (cervical for upper limb and lumbar for lower limb stimulation), with no evidence for responses in the irrelevant electrode grid, thus presenting a spatial double-dissociation. Third, grand-average time-frequency plots delineated responses with a frequency between approximately 50 and 320 Hz at the cervical level and between approximately 50 and 250 Hz at the lumbar level (Fig 3C and 3G), demonstrating the fast nature of these potentials.

Considering recent findings on the complexity of somatosensory processing in the dorsal horn [2], we then went beyond the classical spinal SEPs and assessed whether we could detect responses that occur later than the early N13 or N22 components. Using a cluster-based permutation approach, we did indeed find statistical evidence for such late components: We identified a positive cervical cluster directly after the N13 component (17 to 35 ms, $p = 0.001$; Fig 3D) and two lumbar clusters after the N22 (positive: 28 to 35 ms, $p = 0.002$; negative: 126 to 132 ms, $p = 0.017$; Fig 3H); two out of these three late potentials did also replicate in the independent sample from Experiment 2 (see S1 Text). Taken together, these results provide a comprehensive characterization of spinal SEPs, including responses that occur beyond the initial processing sweep in the spinal cord.

## Enhancing sensitivity via multivariate spatial filtering (Experiment 1)

A main aim of our approach was to enhance the sensitivity for detecting spinal cord SEPs via a multichannel setup and corresponding multivariate spatial filtering analyses, which provide two important benefits. First, multivariate spatial filtering approaches are able to enhance the SNR [41], which is critically important in scenarios such as the low SNR spinal recordings carried out here. Second, by reweighting the multichannel signal on a participant-specific basis, they are able to account for between-participant differences of anatomy and physiology. This

**Table 1. Group-level statistics.**

| SEP/NAP | # | Latency [ms] | Amplitude [µV/a.u.] | SNR | t | p | 95% CI | Cohen's d |
|---|---|---|---|---|---|---|---|---|
| *Mixed median nerve stimulation (hand-mixed)* | | | | | | | | |
| N6 | 32 | 6.22 ± 0.09 | -3.22 ± 0.55 | 14.09 ± 2.3 | -5.89 | <0.001 | [-4.33; -2.11] | -0.98 |
| N9 | 35 | 10.56 ± 0.15 | -2.41 ± 0.21 | 8.8 ± 1.41 | -11.55 | <0.001 | [-2.83; -1.99] | -1.92 |
| N13 (tr) | 36 | 13.25 ± 0.18 | -0.85 ± 0.05 | 9.48 ± 1.16 | -15.75 | <0.001 | [-0.96; -0.74] | -2.63 |
| N13 (vr) | 36 | 13.61 ± 0.17 | -1.40 ± 0.08 | 17.38 ± 3.4 | -17.01 | <0.001 | [-1.56; -1.23] | -2.84 |
| N13 (CCA) | 36 | 13.28 ± 0.17 | -0.47 ± 0.03 | 21.58 ± 2.93 | -16.93 | <0.001 | [-0.53; -0.42] | -2.82 |
| N14 | 30 | 14.30 ± 0.19 | -2.34 ± 0.14 | 24.19 ± 3.04 | -16.95 | <0.001 | [-2.62; -2.06] | -3.09 |
| N20 (CCA) | 36 | 19.81 ± 0.20 | -1.41 ± 0.06 | 23.66 ± 2.41 | -21.85 | <0.001 | [-1.54; -1.28] | -3.64 |
| *Mixed tibial nerve stimulation (foot-mixed)* | | | | | | | | |
| N8 | 34 | 9.28 ± 0.16 | -1.58 ± 0.18 | 10.23 ± 1.72 | -8.64 | <0.001 | [-1.95; -1.21] | -1.44 |
| N22 (tr) | 36 | 23.83 ± 0.29 | -0.80 ± 0.08 | 9.79 ± 1.72 | -9.54 | <0.001 | [-0.97; -0.63] | -1.59 |
| N22 (vr) | 36 | 23.67 ± 0.35 | -0.61 ± 0.06 | 14.14 ± 2.42 | -10.42 | <0.001 | [-0.72; -0.49] | -1.74 |
| N22 (CCA) | 36 | 23.75 ± 0.29 | -0.62 ± 0.06 | 31.28 ± 5.96 | -10.74 | <0.001 | [-0.73; -0.50] | -1.79 |
| N30 | 30 | 32.13 ± 0.43 | -0.53 ± 0.04 | 6.57 ± 1.08 | -13.29 | <0.001 | [-0.61; -0.45] | -2.43 |
| P40 (CCA) | 36 | 40.86 ± 0.38 | 1.42 ± 0.08 | 21.22 ± 2.07 | 18.17 | <0.001 | [1.26; 1.58] | 3.03 |

Descriptive statistics for SEP- and NAP-amplitudes, latencies, and SNR (mean and standard error) and one-sample *t* test of SEP- and NAP-amplitudes in the hand-mixed and foot-mixed conditions of Experiment 1. Note that the brainstem analysis (N14/N30) is based on 30 participants only due to a technical problem (see Methods section).

CCA, canonical correlation analysis; NAP, nerve action potential; SEP, somatosensory evoked potential; SNR, signal-to-noise ratio; tr, thoracic reference; vr, ventral reference; #, number of participants with potentials visible at the individual level.

point is especially relevant in the spinal cord, where our results demonstrate that already at the group level the anatomically defined target channel (red dot in Fig 3B and 3F) does not necessarily capture the strongest deflection of the cervical N13 (slight rostral shift) or the lumbar N22 (slight caudal shift). With individual spatial shifts being even stronger, this indicates a necessity of having a grid of electrodes and correspondingly tailored analyses in order to be able to account for heterogeneity in source location and orientation.

We applied a variant of canonical correlation analysis (CCA) to the preprocessed data of the cervical or lumbar ESG grid, which is a multivariate method that takes information from all sensors of interest into account [40–42]. By finding participant-specific spatial filters that maximize the correlation between two multivariate data sets (here: single SEP trials and the trial-averaged SEP), it computes multiple orthogonal projections, of which we selected the strongest one with a temporal peak at the expected latency and a corresponding spatial pattern with the expected dipole orientation. The resulting group-level cervical N13 and lumbar N22 were similar in shape and latency but clearly exceeded the noise level compared to the single-electrode signal (black traces in Fig 3B and 3G), also resulting in a significantly higher SNR (more than two-fold increase in lumbar data; Table 1 and Fig 4A and 4E), with a large majority of participants showing increased SNR after CCA. Most importantly, the CCA-induced SNR enhancement of the evoked responses allowed for the extraction of cervical and lumbar SEPs at the single-trial level in all participants: Fig 4B–4D and 4F–4H shows single-participant SEPs at the single-trial level, comparing the CCA projected data (right subpanels) with single-electrode data (left subpanels), clearly demonstrating the increase in signal-to-noise level in CCA-cleaned data. This indicates that taking the information from many channels into account provides a fundamental sensitivity increase for detecting even very weak—i.e., trial-wise—spinal responses.

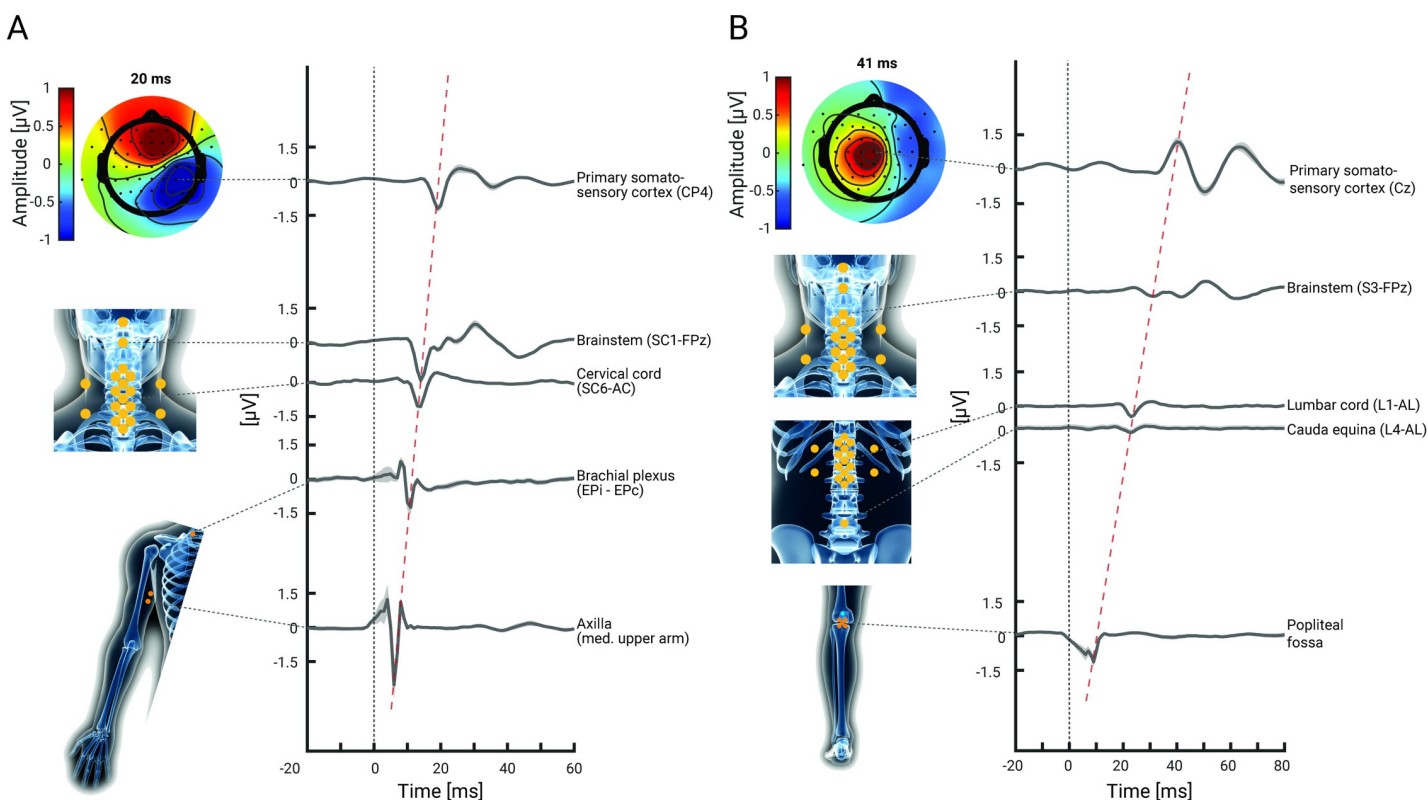

**Fig 2. Grand-average NAPs and SEPs along the somatosensory processing hierarchy.** Group-level responses (*N* = 36) in the hand-mixed (**A**) and the foot-mixed (**B**) conditions of Experiment 1, with shaded error-bands depicting the standard error. The bottom two traces depict peripheral NAPs, the middle trace depicts spinal cord SEPs (referenced ventrally), and the top two traces depict brainstem and cortical SEPs. The grey dashed lines point to the electrode from which the data were obtained, the isopotential plots display the cortical topography, and the red dashed line depicts the temporal progression of the signal along the neural hierarchy. The data underlying this figure can be found in S1 Data.

Furthermore, in order to demonstrate that CCA is not creating artifactual signal due to over-fitting, we carried out a control analysis. More specifically, in each participant we (i) trained CCA on a random selection of 50% of the trials (underlying data: band-pass filtered, anterior-electrode re-referenced, epoched; time windows: 8 to 18 ms for median and 14.5 to 29.5 ms for tibial nerve stimulation); (ii) saved the time-course of the first component; (iii) repeated this procedure a thousand times; and (iv) then calculated all pair-wise absolute correlations between the obtained component time-courses (in the CCA training time-window). This procedure was also carried out on resting-state data, using identical trial timings. At the group level, we then compared the correlation strength between task-based data and resting-state data via a paired *t* test. The main idea of this procedure was to demonstrate that correlations between subsampled CCA components would be substantially stronger in the presence of repeated evoked responses compared to CCA performed on the data from the resting-state data where we do not expect repeated evoked responses. For median nerve stimulation, we obtained a group-average absolute correlation of 0.98 (range across participants: 0.76 to 1) in the task-based data and a group-average absolute correlation of 0.58 (range across participants: 0.46 to 0.73) in the resting-state data; for tibial nerve stimulation, the respective values were 0.96 (range: 0.61 to 1) for task and 0.50 (range: 0.37 to 0.68) for rest. Importantly, component correlations were significantly higher in task-based data than in resting-state data (median nerve stimulation: $t = 27.80$, $p < 9.5 \times 10^{-26}$; tibial nerve stimulation: $t = 25.06$, $p < 3.1 \times 10^{-24}$; one-tailed).

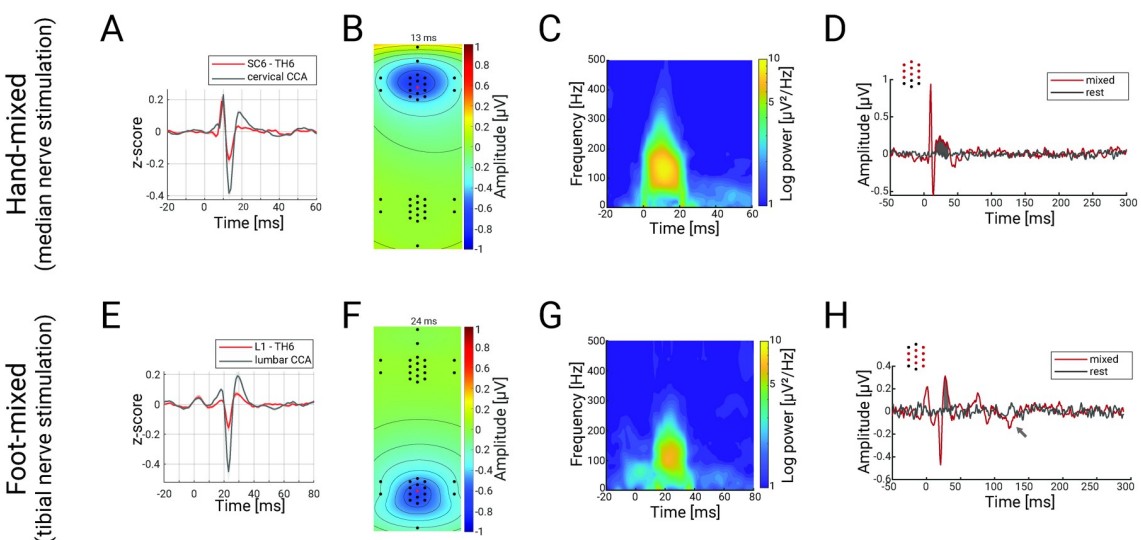

**Fig 3. Spatiotemporal characterization of cervical and lumbar spinal cord potentials.** Panels **A-D** depict responses in the hand-mixed conditions, and panels **E-H** depict responses in the foot-mixed condition. (**A**) and (**E**) Grand-average SEPs across the group obtained from an anatomically defined electrode (hand-mixed: sixth cervical vertebra; foot-mixed: first lumbar vertebra; red trace; both with thoracic reference over the spinous process of the sixth thoracic vertebra (TH6)) or after CCA (black trace), with both signals z-scored for comparison. Note the clear amplitude enhancement of the N13 and N22 after CCA. (**B**) and (**F**) Grand-average isopotential plots (over all spinal channels) in the hand-mixed condition at the peak of the N13 (**B**), and in the foot-mixed condition at the peak of the N22 (**F**). (**C**) and (**G**) Grand-average evoked time-frequency plots in the hand-mixed condition and the foot-mixed condition. (**D**) and (**H**) Results from cluster-based permutation testing for investigating late potentials. Depicted is the grand-average trace over all participants in the stimulation condition (hand-mixed/foot-mixed; red trace) and in simulated epochs from rest data (black trace), averaged over all channels that are part of the identified cluster (displayed as red dots on the top left). The gray areas depict the time-windows with significant differences and the gray arrow indicates an additional significant—but not replicable—potential (see also S1 Fig). The data underlying this figure can be found in S2 Data.

## Detecting spinal SEPs to sensory nerve stimulation (Experiment 2)

Electrical mixed nerve stimulation at the wrist or ankle—as employed in Experiment 1—produces the strongest SEPs in the somatosensory system but is not an ecologically valid type of stimulation (e.g., due to antidromic conduction). To get one step closer toward natural stimulation, in Experiment 2, we additionally stimulated purely sensory nerve fibers of the fingers and toes (for details, see Fig 1). Using this more specific type of stimulation, we did indeed observe clear spinal SEPs, though now with an increased latency (4.3 and 7.6 ms delay for upper and lower limb stimulation, respectively) and reduced amplitude (approximately two-thirds for both upper and lower limb stimulation) compared to mixed nerve stimulation (Fig 5 and S1 Text and S2 Table). Such a pattern of results was similarly observed in peripheral NAPs and cortical SEPs for both finger and toe stimulation (S1 Text and S2 Table) and was also confirmed statistically (S3 Table). Similar to the above-reported mixed nerve results, applying CCA to spinal data resulted in an enhancement of sensory nerve SNR, allowing us to study characteristics of those responses as detailed in the following sections.

A first such example concerned a trial-by-trial investigation of our data (based on fitting linear-mixed-effects (LME) models), assessing whether changes in response amplitude across the processing hierarchy (from peripheral over spinal to cortical levels) would be fully explained by the stimulation condition or whether additional predictive links between the hierarchical levels would be detectable (S1 Text). In brief, we observed that the effects of different stimulation types propagated through the somatosensory processing hierarchy, jointly affecting the amplitudes of peripheral NAPs, spinal cord responses, and initial cortical

**Fig 4. Comparing single-channel SEPs with CCA SEPs.** Panels **A-D** depict responses in the hand-mixed conditions, and panels **E-H** depict responses in the foot-mixed condition. **(A)** and **(E)** SNR for responses obtained from single channels (sixth cervical vertebra [upper] and first lumbar vertebra [lower]) and via CCA; note that the colored lines reflect the SNR of those participants that are displayed in the remaining panels. Panels **(B-D)** and **(F-H)** depict 1,000 single trials of evoked responses (vertical axis) from three representative participants with responses obtained from an anatomically defined electrode shown in the left subpanel and those from CCA shown in the right subpanel; the red arrow indicates the expected SEP latency (hand-mixed: N13; foot-mixed: N22). Note the clear increase in the potentials' single-trial visibility and consistency after CCA. The data underlying this figure can be found in S3 Data. CCA, canonical correlation analysis; SEP, somatosensory evoked potential; SNR, signal-to-noise ratio.

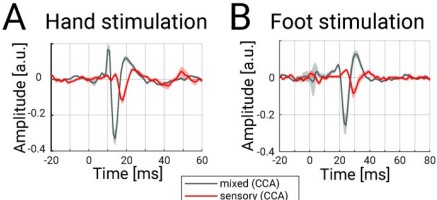

**Fig 5. Spinal SEP to mixed and sensory nerve stimulation.** Depicted is the grand-average over all participants of Experiment 2 in **(A)** the cervical spinal cord to hand-mixed or fingers1&2 stimulation and **(B)** the lumbar spinal cord to foot-mixed or toes1&2 stimulation. All traces were obtained after CCA and the shaded error-bands reflect the standard error (the increased error-band around 0 ms in the lumbar data reflects remaining stimulus artifacts due to imperfect interpolation). The data underlying this figure can be found in S4 Data.

potentials. Interestingly, however, in the foot stimulation condition, additional condition-independent effects of spinal amplitudes on cortical amplitudes were observed, providing first evidence for a trial-by-trial spinocortical link.

## Probing integrative processing along the somatosensory hierarchy (Experiment 2)

Finally, we aimed to study a well-known phenomenon of integration in sensory processing, namely, attenuation or gating effects, which are, for example, observed when stimulating two adjacent fingers: A neuronal response following simultaneous stimulation of both fingers is attenuated compared to the sum of neuronal responses to single finger stimulation. This effect of integrative processing is well studied at the cortical level and has been hypothesized to occur subcortically [43–45], yet unequivocal evidence for such integration occurring already at the spinal level is currently lacking. Therefore, we investigated attenuation effects along the processing hierarchy (i.e., at peripheral, spinal, and cortical levels) and expected (i) that peripheral NAPs would not show attenuation effects (considering that there are no synaptic relays yet); (ii) that cortical SEPs would show such effects (replicating previous observations); and, most importantly, (iii) that the enhanced sensitivity offered by our multichannel spatial filtering approach would allow for uncovering such effects already at the spinal level.

We therefore obtained CCA-extracted amplitudes of cortical and spinal SEPs as well as peripheral NAPs to single-digit and simultaneous digit stimulation. CCA training and component selection was based on mixed nerve data (which have a higher SNR than sensory nerve data), and the chosen spatial filter was then applied to all sensory nerve conditions, ensuring independence of selection and testing. Using these unbiased amplitudes, we assessed the attenuation effect via interaction ratios (IRs): The IR is a measure that quantifies the amplitude reduction of the simultaneous digit stimulation compared to the arithmetic sum of the single-digit stimulations for each participant. Consistent across both upper and lower limb conditions, we obtained clear evidence for attenuation effects not only at the cortical (N20 and P40) but also at the spinal level (N13 and N22); importantly, such effects were not evident at the peripheral level (N6 and N8; Table 2 and Fig 6). While cortical effect sizes of attenuation effects were strongest, spinal effect sizes were already substantial, i.e., in the medium to large range (Cohen's d of 0.5 for lower limb and 1.1 for upper limb). Taken together, our results indicate that robust attenuation effects in somatosensation are not an exclusively cortical phenomenon but already occur at the level of the spinal cord, i.e., after the first synaptic relay.

**Table 2. Group-level IR results.**

| SEP/NAP | IR | tstat | *p* | 95% CI | Cohen's d |
|---|---|---|---|---|---|
| *Hand sensory* | | | | | |
| N6 | -1.83% | -0.60 | 0.56 | [-8.17%; 4.50%] | 0.13 |
| N13 | 20.25% | 5.16 | <0.001 | [12.06%; 28.43%] | 1.13 |
| N20 | 22.21% | 9.03 | <0.001 | [17.12%; 27.30%] | 1.84 |
| *Foot sensory* | | | | | |
| N8 | 6.99% | 0.84 | 0.43 | [-11.28%; 25.27%] | 0.19 |
| N22 | 10.25% | 2.51 | 0.02 | [1.76%; 18.75%] | 0.54 |
| P40 | 26.07% | 6.56 | <0.001 | [17.83%; 34.32%] | 1.37 |

Tested were the IRs of SEPs and peripheral NAPs with a one-sample *t* test.

IR, interaction ratio; NAP, nerve action potential; SEP, somatosensory evoked potential.

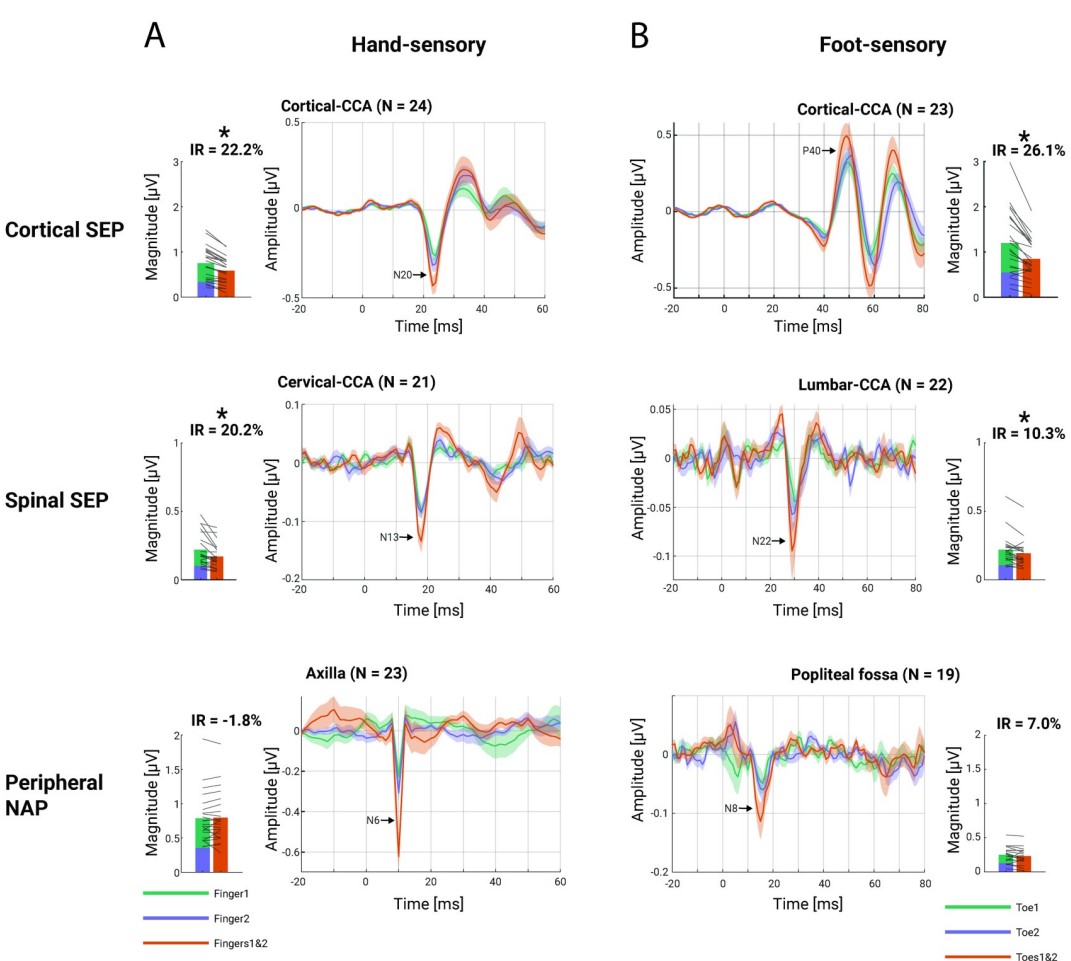

**Fig 6. Attenuation effects along the processing hierarchy.** Potentials following finger stimulation **(A)** and toe stimulation **(B)** from top to bottom: cortical (N20/P40), spinal (N13/N22), and peripheral (N6/N8) responses. The traces in the middle columns display the grand-average response over participants to single-digit stimulation (green and blue traces) and double-digit stimulation (red trace), with the error-band displaying the standard error. The bar plots in the outer columns display the group-average of summed potential amplitudes to single-digit stimulation (green and blue bars) and double-digit stimulation (red bar), with grey lines depicting single-participant data. Note that (i) slightly different numbers of participants entered analyses at the different levels (only those with identifiable and unbiased potentials); (ii) the latency-terminology used here is based on mixed nerve latencies (sensory nerve potentials occur later); and (iii) the scaling of the vertical axes is different between bar-plots and traces (as bar plots depict magnitude data and are based on extracted potential amplitudes at individually optimized latencies). The data underlying this figure can be found in S5 Data.

## Providing a resource for future experiments (Experiments 1 and 2)

Looking ahead, we also aimed to provide a resource for the planning of future experiments by establishing the robustness of the obtained spinal responses. Toward this end, we investigated how many trials are needed to obtain peak amplitudes significantly different from zero at the single-participant level (Fig 7A–7H; left panels) and determined the joint minimal number of trials and participants needed for a significant effect at the group level (Fig 7A–7H; right panels) using resampling approaches.

The most immediately apparent effect is that no matter which outcome is considered, there is a clear order in the level of robustness across the different stimulation conditions, with mixed nerve stimulation giving more robust results than sensory nerve double-stimulation,

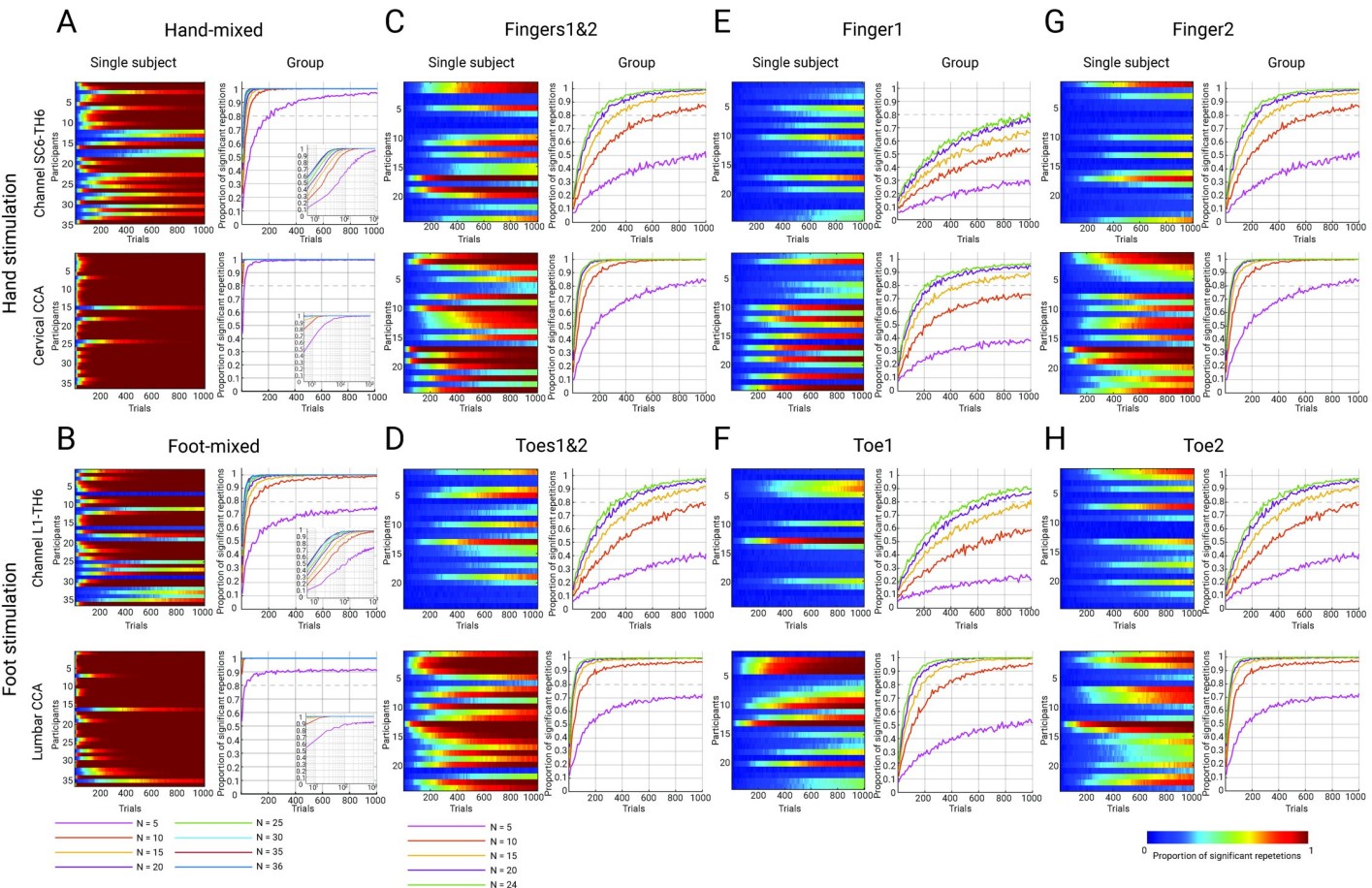

**Fig 7. Robustness of spinal cord SEPs.** Heatmaps display the proportion of significant repetitions for each participant as a function of trial number, and line plots display the proportion of significant repetitions as a joint function of trial number and sample size; data for the anatomically defined target channel are in the top row and for CCA in the bottom row of each panel (insets for mixed nerve stimulation use a logarithmic scale to provide more details). The different conditions are displayed in the following panels: hand-mixed **(A)** and foot-mixed **(B)** (Experiment 1, $N = 36$); hand-sensory (Experiment 2, $N = 24$) with simultaneous finger stimulation **(C)**, and with single finger stimulations **(E and G)**; foot-sensory (Experiment 2, $N = 24$) with simultaneous toe stimulation **(D)**, and with single toe stimulations **(F and H)**; for visual clarity, no more than 1,000 trials are displayed.

which, in turn, leads to more robust potentials than sensory nerve single-stimulation. Thus, whereas in the mixed nerve condition with one target channel, one is almost guaranteed to obtain a significant group-level effect with, e.g., approximately 10 participants and approximately 200 trials (Fig 7A and 7B), many more trials and/or participants would be required in the latter conditions to obtain a significant effect (Fig 7C–7H). Despite this overarching trend, there is, however, also clear interindividual variability in responses (cf. participant #1 and participant #13 in the hand-mixed condition, where approximately 100 versus 1,000 trials were necessary to obtain a significant result in a majority of repetitions).

Another effect that is clearly visible is the beneficial effect of the CCA approach on the robustness of spinal SEPs: In contrast to employing an anatomically defined target channel, employing CCA required smaller numbers of trials to obtain significant results for each participant in a consistent manner (but note that CCA was trained on the entire mixed-nerve data). While this is already visible at the individual-participant and group level in the mixed nerve conditions (Fig 7A and 7B), it becomes even more apparent in the more SNR-limited sensory nerve conditions (Fig 7C–7H). For example, for single-digit stimulation of the index finger

and an anatomically defined target channel (Fig 7E), the use of 24 participants and 1,000 trials was necessary to obtain a significant group-averaged result with a probability of 0.8. In contrast, with the use of CCA (trained on 2,000 trials of mixed-nerve data), either the same number of participants with only approximately 200 trials or 15 participants with approximately 500 trials were already enough to achieve similar results. These results thus allow researchers to make an informed decision on how to set up future experiments in terms of within- and across-participant factors.

### Recording of nociceptive spinal cord responses (Experiment 3)

In a final proof-of-principle experiment ($N = 7$), we aimed to provide an example of the usability of this approach by recording spinal cord—and simultaneously also cortical—responses to nociceptive heat-pain stimulation (induced via a $CO_2$-laser). At the group level, we observed the canonical laser-evoked potentials (LEPs), i.e., the cortical N1 and N2P2 components, with the expected latency (Fig 8A). Most importantly, we also observed a distinct LEP at the spinal level, consisting of a negative deflection at 52 ms (Fig 8B). This response could only be obtained by making use of our multichannel setup and spatial filtering approach, since it could not be detected in single electrode signals. Notably, this group-level response was consistent across data splits (four-fold split depicted in Fig 8C) and observed—with slight latency jitter—in every single participant (Fig 8D).

## Discussion

Here, we report the development of a multichannel electrophysiology approach to noninvasively record spinal cord responses with high precision and sensitivity, incorporating these responses within a comprehensive picture of processing along the somatosensory hierarchy (from peripheral nerves to somatosensory cortex). Across two separate experiments, we provide generalizable results by assessing spatiotemporal response properties in both the cervical and lumbar spinal cord, i.e., the targets of upper and lower limb projections. A central aspect of our approach concerns a reliable extraction and identification of spinal responses in a multivariate way, i.e., reweighting the multichannel signal on a participant-by-participant basis using CCA, which enables single-trial estimation of spinal cord SEPs. In order to allow researchers from various fields to seamlessly build upon our results, we make all data as well as analysis code openly available and also carry out replication and robustness analyses, hoping to provide a status quo of what is currently feasible with multichannel ESG.

### Detailed characterization of spinal cord potentials

Spinal cord SEPs have been studied intensively in the last century, starting with their discovery in humans in an invasive study [46] and followed by noninvasive recordings [20–24]. Here, we employed a novel multichannel approach (including specifically designed multichannel arrays) to expand upon findings from this large body of literature, which encompasses more than 150 publications in healthy humans, but where research had largely subsided.

First, we used a whole-body electrophysiology approach and simultaneously recorded peripheral, spinal, brainstem, and cortical responses to electrical stimulation of a mixed nerve in the upper and lower limbs. This comprehensive recording setup allowed us to embed spinal responses within the temporal progression of the neurophysiological signal along the entire somatosensory processing hierarchy. Second, we compared spinal SEPs following sensory nerve stimulation to those following mixed nerve stimulation and observed reduced peak amplitudes and increased latencies, likely due to the lower number of activated fibers and the additional traveling-distance of nerve impulses, respectively [47,48]. Reassuringly, even with

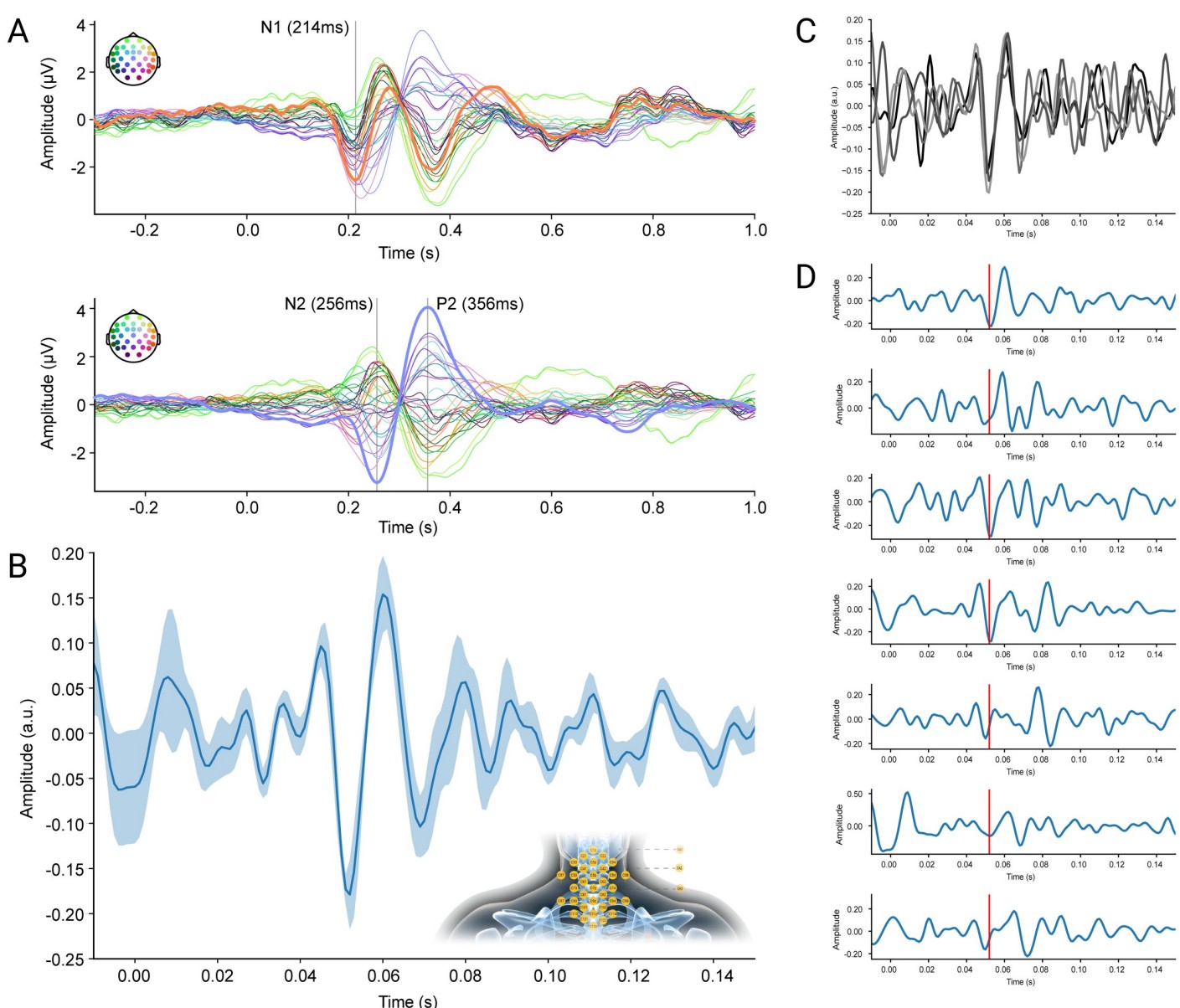

**Fig 8. Cortical and spinal LEPs.** (**A**) Grand-average (*N* = 7) cortical LEPs, obtained from single electrodes and consisting of N1 (latency: 214 ms; data referenced to Cz) in the top row and N2P2 (latency: 256 ms and 356 ms; data referenced to average) in the bottom row; line colors correspond to the electrodes depicted schematically the top left of each panel, with thick lines corresponding to electrodes of interest (N1: T8, N2P2: Cz). (**B**) Grand-average spinal LEP, obtained from CCA and showing the most prominent deflection at 52 ms. The thick line corresponds to the group-average of the first CCA component and the shaded band to the standard error of the mean across participants (amplitudes are in arbitrary units). The electrode patch of Experiment 3 is shown as an inset. (**C**) A four-fold split of trials (after having applied the spatial filter) shows a consistent response in each split at the previously shown latency (amplitudes are in arbitrary units and line colors reflect different splits). (**D**) Single-participant (participants ordered from top (1) to bottom (7)) time-courses of the first CCA component, with the time-course of participant two having been multiplied by −1 (due to the polarity insensitivity of CCA); amplitudes are in arbitrary units. The red line indicates the time-point of maximal deflection observed in the group-level plot. The data underlying this figure can be found in S6 Data.

single-digit stimulation (where only approximately 2,000 to 5,000 nerve fibers can be expected to be activated [49]), we observed mostly large effect sizes, hinting at the potential of our ESG approach to also record responses to ecologically more valid stimulation such as touch, which would be expected to have an even lower SNR. Third, we made use of our multichannel setup

to investigate the spatial distribution of cervical and lumbar SEPs: Both presented as radial dipoles, slightly above the spinous process of vertebra C6 for the N13 and slightly below the spinous process of vertebra L1 for the N22 and with a sagittal center over the cord, speaking against a myogenic origin (which would be expected to result in a more lateralized distribution) [50]. Importantly, our results show a high degree of anatomical plausibility not only by being centered close to the spinal segment of interest but also by showing a spatial double-dissociation: The upper limb N13 is clearly localized in cervical but not lumbar areas, and the lower limb N22 is clearly localized in lumbar but not cervical areas. To our knowledge, such a spatial characterization of spinal SEPs is unique, as even modern MSG-studies are limited to much smaller spatial windows [51,52] (for ESG, see [27]) and would thus not allow for such insights. Obviously, our data do not allow pinpointing the origin of these potentials within the spinal cord gray matter, but their postsynaptic nature has been established [32,53] and animal work suggests that they are generated by deep dorsal horn interneurons [54–56], likely as part of the postsynaptic dorsal column (PSDC) pathway, which is a prominent source of input to the dorsal column nuclei [57–59].

Finally, based on the recent evidence for extensive processing of afferent signals within the dorsal horn [2,3], we investigated the existence of late spinal potentials and indeed observed such SEP components following the cervical N13 (17 to 35 ms) and the lumbar N22 (28 to 35 ms). Similar late spinal potentials had been descriptively mentioned as part of a triphasic wave in some of the earliest invasive and noninvasive spinal recordings [20,21,60,61], but here we provide firm statistical evidence for their existence at the group level for the first time. With respect to the origin of these late potentials, a myogenic source has been ruled out [61] and a contribution from late top-down brainstem potentials [44] is unlikely given their lumbar presence, rather pointing toward a local spinal origin, with a possible neurophysiological mechanism being primary afferent depolarization [56]. We further obtained tentative evidence for an ultralate negative lumbar potential following lower limb stimulation after >100 ms. To our knowledge, no spinal SEPs have hitherto been reported at such latencies, although there are hints from early neuromagnetic neck recordings [62] and recent spinal recordings based on optically pumped magnetometers [63] have also shown late spinal evoked fields. Taken together, the general possibility to detect late potentials opens the door for investigating local spinal processing going beyond a simple relay of information [2] as well as supraspinal modulatory influences on processing in the dorsal horn [64], and here the millisecond resolution of our approach will be ideally suited to disentangle top-down from bottom-up effects.

## Enhanced SNR and single-trial responses via multivariate spatial filtering

Traditionally, the analysis of SEPs from *ESG* data is based on acquiring a large number of trials, with most studies using single or very few spinal electrodes (though there are a few exceptions [21,28,65]) and then analyzing *single-channel* data. Conversely, methodological advances in *EEG* data acquisition and analysis now allow for a better separation of signal from noise and use high-density *multichannel* montages for construction of spatial maps, in which the data of the whole set of electroencephalography (EEG) electrodes are treated as a multivariate signal [66–69].

Our high-density ESG-montage thus enabled the application of methods that combine the information from many channels via spatial filters. Specifically, we used a CCA-based approach—that has previously been applied for extraction of early cortical SEPs [40–42,70,71] —and show that spinal SEP extraction is markedly improved with such a multichannel spatial filtering approach. We believe this approach is especially beneficial for spinal data for two reasons. First, the ESG signal is particularly affected by physiological noise from cardiac and

myogenic sources [21], leading to a low SNR with single-trial amplitudes usually hidden in background noise. Second, despite substantial interindividual differences in the location of spinal segments relative to vertebrae [72,73], the latter are used as anatomical landmarks for electrode placement. A spatial filter that compensates for such interindividual differences will be beneficial for group-level analyses, but also for recovering signals in individual participants, where an electrode placed on a specific anatomical landmark might not capture the spatial peak of the response. Finally, because of volume conduction, potentials from the activation of spinal sources will be reflected in many electrodes, and, thus, a simultaneous use of the signals from many electrodes may also potentially allow extraction of multiple sources when using multivariate methods.

By improving the SNR of ESG data, our spatial filtering approach allows not only for extracting more robust spinal SEPs but also for studying the variability in spinal SEP amplitudes at the single-trial level. While there are many benefits to this approach (see below), here we employed it to assess how trial-by-trial response amplitudes cofluctuate across different processing levels. We observed that the effects of different stimulation conditions (i.e., single-digit, double-digit, and mixed nerve) corresponded to shared variance across the somatosensory processing hierarchy, encompassing peripheral NAPs, spinal SEPs, and early cortical SEPs. This covariance presumably reflected the number of stimulated nerve fibers (which varied between stimulation conditions) as well as the internal state of the activated neuronal populations. Yet, additional condition-independent variations might be worth further investigation: During foot stimulation, spinal responses predicted cortical responses, providing a neurophysiological spinocortical link on the single-trial level.

## Integration effects are present already at the spinal level

Most importantly, we assessed a fundamental question of sensory processing, namely, at which levels of the processing hierarchy information from the receptors is integrated, by testing for integrative processes at peripheral, spinal, and cortical levels. In order to do so in a robust manner, we used CCA to extract SEP amplitudes to single-digit and double-digit stimulation and quantified the attenuation effect—a reduced response to double-digit stimulation compared to the summed-up responses to single-digit stimulation—as a measure of integration. Integration effects were not evident in the peripheral nervous system, where response amplitudes faithfully reflected the applied stimulation. Conversely, significant integration effects with medium to large effect sizes were evident consistently after the first synaptic relay, i.e., not only in cortex but already in the spinal cord, the first processing station in the central nervous system. The cortical findings are in line with several previous studies [43,45,74], but the robust spinal results—which were observed for both upper and lower limb stimulation—go far beyond the previous literature, where only anecdotal evidence of such effects existed at the cervical level [75,76]. While the simultaneous recording and assessment of integration effects at peripheral, spinal, and cortical levels is a first, to our knowledge, the observed progression of increasingly stronger integration effects along the neural hierarchy has been suggested to be a consequence of increasing receptive field size [44].

Two mechanisms have been discussed to underlie integration effects: occlusion and lateral inhibition [74,76,77]. Either mechanism could be at work in the spinal cord, considering the integrative nature of many deep dorsal horn interneurons [2,3] as well as the receptive-field organization of wide dynamic range neurons [78], both of which have been suggested to contribute to the observed spinal SEPs [35,36,54,55]. Future work using experimental designs tailored to dissociate these two mechanisms [74] might help to shed more light on the underlying processes at the spinal level.

## Assessment of nociceptive spinal cord responses

In a final experiment, we demonstrated the versatility of our noninvasive approach by providing corticospinal electrophysiological recordings of responses to nociceptive stimulation in a heat-pain paradigm. While there is a multitude of EEG studies assessing cortical responses to various forms of nociceptive stimuli (e.g., LEPs [79]; contact heat-evoked potentials (CHEPs [80]); pinprick-evoked potentials (PEPs [81])), a noninvasive assessment of spinal responses to any type of nociceptive stimulation has hitherto not been reported to our knowledge. In a first proof-of-principle experiment, we therefore leveraged the sensitivity increase afforded by our multichannel setup to demonstrate concurrently recorded cortical and spinal LEPs. The spinal response occurred at a time point consistent with the activation of nociceptive A-delta fibers (i.e., in an early latency-range of what would be expected based on microneurographic single-fiber recordings [82]) and was observable to varying degrees in every single participant after spatial filtering.

While this first demonstration of noninvasively recorded spinal LEPs obviously awaits replication (and might benefit from more sophisticated time-window selection for CCA training, possibly informed by recent computational heat-transfer models [83]), it is a promising step to investigate entire CNS mechanisms underlying the experience of pain in health and disease. A noninvasive and direct window into spinal nociceptive processing is highly relevant for pain research, considering that the spinal cord is not only the first CNS processing station for nociceptive stimuli [84] but also a target of powerful descending control mechanisms [85] and a structure often implicated in pain chronification [86].

## Insights for future electrospinography experiments

One outstanding question is how the advances introduced by our approach might benefit other fields of human neuroscience, i.e., inspire new work on spinal cord function outside the domain of somatosensation. An immediate experimental implication arises from the here-developed denoising approach: While it was recognized early on that cardiac artifacts dominate the ESG signal [21] and that massive trial-averaging or cardiac-gating was thus necessary, we instead achieved a direct removal of the cardiac artifact via a denoising algorithm [87], eliminating these previous limitations. This allows, for example, to deliver stimuli spaced across the cardiac cycle, and we are thus envisioning the use of our approach for investigating interoceptive processes, where spinal pathways are of importance for brain–body communication but not yet studied [88,89]. In addition, the sensitivity increase afforded by our multichannel approach in combination with spatial filters is of benefit for domains where massive trial-averaging is impossible (e.g., in pain research due to ethical and safety reasons) or for experimental paradigms where only a few or even single trials are of interest (e.g., in deviance-detection designs).

Our approach could also provide clinical insights, considering that spinal pathologies are a core part of many neurological disorders, such as multiple sclerosis [6], spinal cord injury [5], or chronic neuropathic pain [7]. Relating to this, great strides have recently been made in the recovery of function after spinal cord injury and stroke via spinal neurostimulation [8,9,90,91]. Here, a noninvasive and temporally resolved window on such processes—as provided by the CCA-enabled single-trial sensitivity—might offer mechanistic insights into processes underlying such recovery, especially considering the role of afferent input in the treatment of these maladies, as successfully characterized by our approach. Similarly, there are multiple initiatives aimed at developing biomarkers for analgesic drug development to target chronic pain [10,11,92,93] and considering that alterations in spinal processing are assumed to be a core feature of chronic pain development and maintenance [94–96], sensitive spinal

recordings would be very helpful. In such endeavors, spinal SEPs could potentially serve as objective, noninvasive, and innocuous biomarkers [35,36].

In any case, considering that underpowered studies are a troubling issue in neuroscience [97], both experimental and clinical studies that could arise from this work would need to be well powered. In order to facilitate the planning of such studies, we provide group-level effect sizes, which—reassuringly—were similar across both experiments and mostly in the large range. In addition, we used resampling approaches on both data sets to (i) estimate the minimal number of stimuli to obtain a significant result at the participant level and (ii) jointly estimate the minimal number of stimuli and participants to obtain a significant result at the group level. Simulating experiments this way allows for giving specific recommendations, such as that for mixed nerve stimulation acquiring approximately 200 trials in approximately 10 participants with single-channel recordings almost guarantees a significant group-level effect.

Finally, we hope that our noninvasive approach in humans will provide a macroscale complement to research in animal models, where invasive recording techniques—such as multielectrode recordings [98] or calcium imaging [99]—allow detailed and mechanistic insights into spinal processes occurring at the micro- and mesoscale. It is important to note that our approach of not only recording cervical but also lumbar spinal cord responses could provide a unique across-species bridge, considering that the vast majority of spinal recordings in experimental animal models are carried out in the lumbar cord.

## Limitations and comparison with other CNS-neuroimaging approaches

There are several limitations of our approach that are worth discussing. First, the supine positioning of participants might have led to a higher noise level in the ESG data due to electrode movements. While there are several alternative positions, we decided to record data in supine position based on extensive piloting, in which this position was reported to offer the most comfort over the course of the experiment without degrading data quality (e.g., due to tonic muscle activity). Second, we had hoped to reliably record brainstem SEPs arising from the cuneate nucleus (N14 [100]) and gracile nucleus (N30 [101]), as these are direct recipients of output from the spinal cord via the PSDC pathways [58]. Despite using optimal signal extraction leads, observing brainstem potentials was not possible in all conditions, mainly due to the limited SNR to digit stimulation. Third, it is important to point out that this study introduced a novel methodological approach and was thus focused on the detection of spinal responses to carefully controlled stimulation that gives rise to a strongly synchronized high-amplitude signal. One might therefore ask whether this method will perform well under more naturalistic conditions, such as mechanical or thermal stimulation. We believe that the combination of methodological improvements introduced here should also be helpful in such low-SNR scenarios, as already demonstrated exemplarily for single-digit stimulation. Finally, it should also be noted that there is some loss of objectivity when using CCA, considering that a priori knowledge informed the time-period for training CCA and the choice of component—this might be alleviated in the future by developing automated procedures based on predefined criteria.

In terms of comparison with other neuroimaging methods for assessing the entire CNS, we note that only fMRI and MEG (based on optically pumped magnetometers (OPMs) [102,103]) have so far been used for simultaneous assessment of corticospinal processes. While corticospinal fMRI—as employed for studying the interactions between supraspinal and spinal structures that underlie resting-state connectivity [104,105], motor control [106,107], or top-down modulation of nociceptive processing [108,109]—offers unparalleled spatial resolution, it is an indirect measure of neuronal processes with ensuing low temporal resolution. Here, our

approach would provide an important complementary assessment, as it would, for example, allow for a temporally precise delineation of possible interactions between top-down and bottom-up responses at the spinal level due to its millisecond resolution. OPM-MEG has recently been employed in a proof-of-principle study to simultaneously record spinal and cortical somatosensory-evoked responses similar to those investigated here in Experiment 1 [63] (see [110] for corticospinal recordings during a motor task). While the high costs and limited bandwidth of many OPM sensor types currently limit widespread adoption (especially when interested in very fast spinal responses as investigated here), the wearable nature and flexible arrangement possibilities of OPM-MEG make this a very promising methodological approach for entire CNS assessments with high temporal precision.

### Outlook

In conclusion, we established an approach for the noninvasive recording of spinal cord responses that should be readily accessible and widely available, addressing a previously missing link in the study of reciprocal brain–body communication. Our method provides direct recordings of electrophysiological responses with high temporal precision (allowing to investigate different response components, i.e., early and late potentials), has a high sensitivity due to the multivariate combination of spinal multichannel data (enabling single-trial estimates), and is integrated with the recording of afferent and efferent signals (peripheral and supraspinal responses). We believe that this approach could be extended to other settings of natural stimulation—such as social touch or pain (for which we provide initial evidence)—and is not only suitable for investigating hard-wired bottom-up processing but also its modulation by various factors, such as signal integration as demonstrated here to already take place in the human spinal cord. We thus hope to have provided a comprehensive approach that allows for a sensitive and direct assessment of spinal cord responses at millisecond timescale in various fields beyond somatosensation and anticipate its use in the context of interrogating the spinal cord's role in the interplay of bottom-up and top-down processes that together give rise to our sensations in health and disease.

## Materials and methods

### Participants

**Experiment 1.** A total of 42 healthy right-handed volunteers participated in this experiment. Two participants were not able to successfully complete the experiment (cigarette craving in one case, bathroom use in another case), and their data were thus discarded. Four participants were excluded due to absent peripheral potentials, leading to a final sample size of 36 participants (18 female; age: 25.5 ± 3.5 years (mean ± SD)). All participants provided written informed consent, and the study was approved by the Ethics Committee at the Medical Faculty of the University of Leipzig. Please note that the final sample-size of 36 participants was specified in a preregistration prior to the start of the study and was chosen in order to detect a medium-sized effect (Cohen's d = 0.5) with a power of 90% (at an alpha level of 0.05 with one-tailed testing).

**Experiment 2.** A total of 26 healthy right-handed volunteers participated in this experiment. Two participants were excluded due to absent peripheral potentials in the mixed nerve stimulation condition, leading to a final sample size of 24 participants (12 female; age: 24 ± 4.5 years (mean ± SD)). All participants provided written informed consent, and the study was approved by the Ethics Committee at the Medical Faculty of the University of Leipzig. Please note that the final sample size of 24 participants was specified in a preregistration prior to the start of the study. This was based on a power calculation of data from of the 36 participants in

Experiment 1, where we observed an effect size of $d = -0.85$ for median mixed nerve stimulation and of $d = -0.62$ for tibial mixed nerve stimulation (in 30 Hz high-pass–filtered, but otherwise uncleaned, data). Taking the smaller of these two effect sizes, and aiming for a power of 90% (at an alpha level of 0.05 with one-tailed testing) resulted in a necessary sample size of 24 participants. Although we were using results obtained from mixed nerve stimulation as the basis for our power calculation (which is known to result in stronger responses than those from stimulation of a purely sensory nerve), we employed a conservative way to estimate our effect size: (i) we used raw data that were only preprocessed by a high-pass filter; (ii) we based our power calculation on the lumbar potential that is possibly more difficult to detect; and (iii) we selected the same electrode in each participant (cervical: SC6, lumbar: L1) to calculate the group statistics, which is rather conservative especially for the lumbar channels, because the location of the lumbar segments of the spinal cord differs extensively between participants [73].

## Experimental design

We conducted two experiments in which human participants received electrical stimuli to mixed or sensory parts of an arm and of a leg nerve. In Experiment 1, only mixed fibers were stimulated, specifically of the median nerve at left wrist and of the tibial nerve at the left ankle. In Experiment 2, the same mixed nerve stimulation was applied, and additionally sensory parts of the nerves were stimulated (two fingers or two toes). In both experiments, electrophysiological signals were recorded at different levels of the processing hierarchy—at the peripheral nerve, the lumbar and cervical spinal cord, the brainstem, and the cortex.

**Experiment 1.** The experiment had a repeated-measures design, meaning that each participant underwent all experimental conditions. The experiment consisted of two conditions, named hand-mixed and foot-mixed in the following. In the hand-mixed condition, the left hand of the participant was stimulated with electrical pulses to the median nerve at the wrist. In the foot-mixed condition, the left foot of the participant was stimulated with electrical pulses to the posterior tibial nerve at the ankle. We refer to these conditions as "mixed," because at the wrist and the ankle, the median and tibial nerve, respectively, are mixed nerves, i.e., contain both sensory and motor nerve fibers. Fig 1A displays the experimental timeline of Experiment 1.

**Experiment 2.** Similar to Experiment 1, this experiment also had a repeated-measures design, though now consisting of eight conditions, named hand-mixed, finger1, finger2, fingers1&2, foot-mixed, toe1, toe2, and toes1&2. The hand-mixed and foot-mixed conditions were the same as in Experiment 1 (except for differences in the interstimulus interval and being presented completely in one block each). In the finger stimulation conditions, the index and middle finger of the participant's left hand were stimulated with electrical pulses. These pulses could occur in three different ways: to the index finger only (finger1), to the middle finger only (finger2), or to both fingers simultaneously (fingers1&2). In the toe stimulation conditions, the first and second toe of the participant's left foot were stimulated with electrical pulses either to the first toe only (toe1), to the second toe only (toe2), or to both toes simultaneously (toes1&2). We refer to all finger and all toe stimulation conditions also as "hand-sensory" and "foot-sensory" conditions, because at the fingers and the toes, the median and the stimulated branches of the posterior tibial nerve contain only sensory nerve fibers. Fig 1B displays the experimental timeline of Experiment 2.

## Electrical stimulation

**Experiment 1.** The electrical stimulus was a 0.2-ms square-wave pulse delivered by 2 constant-current stimulators ("DS7A", Digitimer, Hertfordshire, United Kingdom; one stimulator

for each nerve) via a bipolar stimulation electrode with 25 mm electrode distance ("reusable bipolar stimulating surface electrode," Spes Medica, Genova, Italy) to the left median or the left posterior tibial nerve, respectively. The stimulation electrodes were placed (with the cathode being proximal) at the palmar side of the wrist (median nerve stimulation) and at the median side of the ankle (posterior tibial nerve stimulation). The stimulation intensity was set to just above the individual motor threshold, which was defined as the intensity at which a participant's thumb or first toe started to twitch (visually determined). All participants perceived the stimulation intensity as a distinct, but not painful, sensation.

**Experiment 2.** Equipment and electrode placement for mixed nerve stimulation was identical to what is described above for Experiment 1. For finger or toe stimulation, ring electrodes ("digital electrode for recording and stimulation," Spes Medica, Genova, Italy) were attached with the cathode being proximal to participants' left index finger and left middle finger as well as left first toe and left second toe. While we intended to stimulate mixed and sensory parts of the same nerve, when stimulating the fingers or toes, it is not possible to clearly differentiate which nerve is stimulated, since there is an individual variability in the spatial distribution of the dermatomes [111,112]. Therefore, it is important to keep in mind when interpreting our results that during stimulation of the index and middle finger, sensory fibers of the median as well as the ulnar and radial nerve might be stimulated (lower limb: sensory fibers of the superficial and deep peroneal nerves). Each of the fingers or toes were stimulated by a different stimulator. The stimulation intensity was set to three times the detection threshold, which was determined via the method of limits. If necessary, i.e., if participants reported to experience the stimulus as less intense over time, the stimulation intensity was slightly increased in-between stimulation blocks based on experience from pilot experiments as well as suggestions by earlier work [113]. The applied intensity was never perceived as being painful.

## Electrographic recordings

**Experiment 1.** All electrographic signals were recorded with TMS-suitable Ag/AgCl electrodes ("TMS-compatible multitrodes," Easycap GmbH, Herrsching, Germany). For EEG, 64 electrodes were arranged on an EEG cap (Easycap GmbH) with standard positions according to the 10–10 system and referenced to the right mastoid (RM). Recorded EEG-channels were the following: Fp1, Fp2, F3, F4, C3, C4, P3, P4, O1, O2, F7, F8, T7, T8, P7, P8, AFz, FCz, Cz, Pz, FC1, FC2, CP1, CP2, FC5, FC6, CP5, CP6, FT9, FT10, LM (left mastoid), Fz, F1, F2, C1, C2, AF3, AF4, FC3, FC4, CP3, CP4, PO3, PO4, F5, F6, C5, C6, P5, P6, AF7, AF8, FT7, FT8, TP7, TP8, PO7, PO8, FPz, CPz, F9, and F10. An active ground electrode was placed at POz.

For ESG, 39 electrodes were placed on the upper body, with the largest part of the electrodes placed into one cervical and one lumbar electrode patch. These patches were custom-made and consisted of the same fabric used for the EEG cap (kindly provided by Easycap GmbH). ESG data were referenced to an electrode positioned over the spinous process of the sixth thoracic vertebra (TH6), and the following electrodes were located at anatomical positions: electrode SC1 at the first cervical vertebra, electrode SC6 at the spinous process of the sixth cervical vertebra, electrode L1 at the spinous process of the first lumbar vertebra, and electrode L4 at the spinous process of the fourth lumbar vertebra. An additional 16 electrodes were organized in a grid around each one of the two spinal target electrodes SC6 and L1 (Fig 1). The grid organization, which was developed in pilot experiments, aimed at capturing the spatial distribution of the spinal signal. The midline of this grid was positioned vertically on the spine and consisted of five electrodes (the third one being the spinal target electrode) with a vertical interelectrode distance of 2 cm. Two further vertical lines of four electrodes each were placed 1 cm to the right and left of the midline electrodes and another two vertical lines of two

electrodes each were placed 5 cm to the right and left of the midline. In addition to these dorsally placed electrodes, there were two ventrally placed electrodes—one supraglottic (AC) and one supraumbilical electrode (AL). Such ventral electrodes have been described to be beneficial for SEP extraction in the literature [26,27,114,115]. Because the EEG and ESG montage used different references, we added Fz to both montages with channel name "Fz" in the EEG montage and "Fz-TH6" in the ESG montage, as this allows to combine the two montages into one by re-referencing at a later point. In six out of the 36 participants (sub-001 to sub-006), Fz-TH6 was missing in the ESG setup due to a technical error. The active ground electrode stabilized the signal via the "driven right leg" principle. It was placed at POz in the EEG montage and in the middle between TH6 and S20 in the ESG montage. Please see also our reasoning regarding the placement of the spinal reference in S1 Text.

In addition to EEG and ESG, we also recorded several other types of data. First, electroneurographic (ENG) data—i.e., peripheral NAPs—of the median nerve were recorded at the level of the left axilla (over the biceps, reference electrode proximal, distance 3 cm between electrodes) and the left Erb's point (referenced to right Erb's point). Peripheral NAPs of the posterior tibial nerve were recorded from the popliteal fossa (with five electrodes: one electrode was placed in the center of the fossa and four electrodes around it at a distance of 1 cm; all knee channels were referenced to a 3-cm proximal electrode). Second, electrocardiographic (ECG) data were recorded from an electrode placed at the left lower costal arch and referenced to a right subclavicular electrode. Third, electromyographic (EMG) data were recorded at the hand from the abductor pollicis brevis muscle and at the foot from the flexor hallucis brevis muscle, with the EMG electrode being placed over the muscle belly and the reference electrode being proximal (please note that EMG data are not reported in this manuscript). Fourth, we recorded the participants' respiratory activity (with a respiration belt: "reusable respiratory effort sensor," Spes Medica S.r.l., Genova, Italy; data also not reported here).

We aimed at keeping impedances at all electrodes below 10 kOhm. All electrographic signals were recorded with NeurOne Tesla amplifiers and software (Bittum Corporation, Oulu, Finland), applying an anti-aliasing filter at 2,500 Hz with a lower cutoff at 0.16 Hz and sampled at a rate of 10,000 Hz.

**Experiment 2.** The employed recording equipment as well as the ESG, ECG, and ENG electrode placement was identical to what is described above for Experiment 1. EEG was recorded using 39 electrodes arranged on an EEG cap with standard positions according to the 10–10 system and referenced to the RM. Recorded EEG-channels were the following: Fp1, Fp2, F3, F4, C3, C4, P3, P4, O1, O2, F7, F8, T7, T8, P7, P8, AFz, Fz, Cz, Pz, FC1, FC2, CP1, CP2, FC5, FC6, CP5, CP6, LM, FCz, C1, C2, FC3, FC4, CP3, CP4, C5, C6, and CPz. The electrooculogram was placed lateral to the outer canthi (EOGH) and in the center below (EOGV) the right eye and used the same reference as EEG. An active ground electrode was placed at POz. EMG was not recorded in this experiment.

## Experimental procedure

**Experiment 1.** First, the EEG, ESG, ENG, EMG, and ECG electrodes were attached to the participant's skin. Next, the respiration belt was attached at the level of the ninth/tenth rib. Then participants were asked to lay down on a cushioned bench on their back in a semidarkened and acoustically shielded EEG-cabin. For participant comfort, the head support of the bench was slightly raised and a cushion roll was placed under their knees. Next, electrical stimulation location and intensity were determined and participants were instructed to look at a fixation cross during the stimulation blocks, which was attached to the ceiling. The experiment started with 5 minutes of resting-state recording (eyes open) followed by eight stimulation

blocks, each consisting of 500 stimuli. During one block, stimuli were delivered to one nerve only, i.e., either the median or the posterior tibial nerve (thus, there were four median and four posterior tibial nerve stimulation blocks in total). The stimulation blocks were presented in alternating order, and the order was counterbalanced across participants. Another two blocks of similar length followed at the end of the experiment—these are not discussed here as they were part of another project and are thus explained in further detail elsewhere [71]. We used an interstimulus interval of 763 ms with a uniformly distributed jitter of +/− 50 ms in steps of 1 ms. Taken together, each nerve received 2,000 stimuli overall. The experiment took approximately 5.5 to 6 hours, with the presentation of the experimental stimulation blocks (including breaks) taking approximately 90 minutes.

**Experiment 2.** Since the attachment of the recording equipment to the participants and the instruction of the participants were identical to Experiment 1, in the following, we only list details specific to Experiment 2. Before each experimental block started, the individual stimulation intensity was adjusted if necessary. The experiment started with 5 minutes of resting-state recording followed by 10 stimulation blocks (with short breaks between blocks). There were four different types of stimulation: (i) mixed nerve stimulation of the median nerve (one block); (ii) mixed nerve stimulation of the tibial nerve (one block); (iii) sensory nerve stimulation at the fingers (four blocks); and (iv) sensory nerve stimulation at the toes (four blocks). All blocks of one stimulation type were presented in a row (with pauses between blocks), but the order in which the four stimulation types were presented was balanced across participants. There was one block for hand-mixed and one block for foot-mixed stimulation, and each of these blocks contained 2,000 stimuli. Sensory nerve stimulation was separated into four blocks (1,500 stimuli each) of finger and four blocks (1,500 stimuli each) of toe stimulation. During each finger stimulation block, finger1, finger2, and fingers1&2 were stimulated in a pseudo-random order, such that each of the three stimulation conditions occurred 500 times. The same procedure was employed for the toe stimulation blocks, with the only difference that toe1, toe2, and toe12 were stimulated in pseudorandom order. Each type of digit stimulation (finger1/toe1, finger2/toe2, fingers1&2/ toes12) thus consisted of 2,000 stimuli. All stimuli were delivered with an interstimulus interval of 257 ms with a uniformly distributed jitter of +/− 20 ms in steps of 1 ms. The experiment took approximately 6 to 6.5 hours, with the presentation of the experimental blocks (including breaks) taking approximately 90 minutes.

## Data processing and statistical analysis (Experiment 1)

Unless noted otherwise, all data were analyzed using MATLAB R2019b (The MathWorks, Natick, Massachusetts, United States of America) and the EEGlab toolbox [116].

**Stimulation artifact removal.** Electrical stimulation of peripheral nerves as employed here induces an artifact in all channels at the time point of stimulation and was removed by interpolation (using a piecewise cubic hermite interpolating polynomial). Since the temporal spread of this artifact differed among participants, as well as in cervical and lumbar channels, we defined individual artifact windows for cervical and lumbar levels by finding the beginning and the end of the artifact in the average over all trials and all cervical or lumbar ESG channels. At the cervical level, average artifact windows ranged from −1.8 ms (SD = 0.8 ms) to 4.4 ms (SD = 1.4 ms) and at the lumbar level from −2.9 ms (SD = 1.4 ms) to 7.1 ms (SD = 2.8 ms).

**EEG data preprocessing.** First, the stimulation artifact was interpolated using the previously identified cervical artifact windows and the continuous EEG signal was down-sampled to 1,000 Hz (anti-aliasing filter with cutoff at 0.9 and transition bandwidth at 0.2). Second, artifact sources were identified in the signals using ICA. For this, overly noisy channels were removed from the signal—based on visual inspection of the power spectral density and the

trial-based root mean square activity in each channel—and interpolated (this was the case for one channel in five participants). Zero-phase IIR filtering was then applied to the continuous concatenated signal from all stimulation blocks (i.e., median and tibial nerve stimulation), consisting of a high-pass filter at 0.5 Hz and a low-pass filtered at 45 Hz (Butterworth, fourth order). On the filtered signal, independent component analysis (ICA; Infomax [117]) was performed and ICs reflecting eye blink, heart, and muscle artifacts were identified. Third, ICs identified as representing artifactual sources were removed from the EEG signal preprocessed in the same ways as for ICA, with the difference that it (i) consisted of concatenated blocks of each stimulation condition only (i.e., hand-mixed or foot-mixed) and (ii) was zero-phase IIR filtered with a notch (48 to 53 Hz) and a band-pass (30 to 400 Hz) Butterworth filter of fourth order. Fourth, the ICA-cleaned signal was re-referenced to average reference, and remaining noisy time points were identified in lower frequencies (1 to 15 Hz) using a threshold of five standard deviations and in higher frequencies (15 to 45 Hz) using a threshold of 60 µV. If more than 50% time points were identified in one channel, this channel was removed from the data and interpolated. In one participant, seven channels were removed from the hand-mixed condition, and in another participant, 18 channels were removed from the foot-mixed condition. Fifth, the cleaned signal was cut into epochs from 200 ms before to 700 ms after stimulus onset and baseline-corrected (with a reference interval from −110 ms to −10 ms before stimulus onset). In the hand-mixed condition, this procedure led to an average of 97.9% remaining trials (range across participants: 886 trials to 2,000 trials) and in the foot-mixed condition to an average of 97.5% remaining trials (range across participants: 992 trials to 2,000 trials).

**ESG data preprocessing.** After the stimulation artifact was interpolated in the individually defined cervical and lumbar artifact windows, the ESG data were down-sampled to 1,000 Hz.

Since ESG data are known to present with severe cardiac artifacts [21], we aimed to correct for these. In each participant, we therefore first identified R-peaks in the ECG channel using an automatic procedure provided by the FMRIB plugin for EEGlab (https://fsl.fmrib.ox.ac.uk/eeglab/fmribplugin/), which was followed by visual inspection and manual correction if necessary. Next, the heart artifact was removed from each ESG channel separately, using an approach that is a modification of a method previously developed for removing ballistocardiographic artifacts in simultaneous EEG-fMRI recordings [87]. First, a principal component analysis (PCA) was applied to a matrix of all heart artifacts (artifact × time) in one channel, with the time window of each heart artifact ranging from −0.5 * median(RR) to +0.5 * median(RR) around each R-peak (with RR referring to the interval between R-peaks, i.e., the heart-period). Then, an optimal basis set (OBS) was created based on the mean heart artifact and the first 4 components obtained from the PCA. Finally, this OBS was fitted to each heart artifact and then removed from it.

After correction for cardiac artifacts, noisy channels were identified via visual inspection of the power spectral density and one channel in five participants was removed (no interpolation of missing channels was performed at the spinal level).

The analysis steps described below were performed in the concatenated blocks of one condition (rest, hand-mixed or foot-mixed) and, because we wanted to investigate SEPs with different references, were carried out separately for differently referenced data sets. In addition to the recording reference located over the spinous process of the sixth thoracic vertebra (TH6), we also made use of a ventrally located reference, because it has been reported that this can be beneficial for SEP extraction [26,114]—the ventral reference was channel AC in the hand-mixed and channel AL in the foot-mixed condition. First, a zero-phase IIR filtering was applied to the data with a notch (48 to 53 Hz) and a band-pass (30 to 400 Hz) Butterworth filter (fourth order). Second, time points with absolute ESG activity above 100 µV were removed

from the continuous data. If in one channel more than 50% of time points were identified, the whole channel was excluded instead. No further channels were removed, and together with the channel exclusion based on the spectrum in the whole sample, an average of 0.1 channels were removed (SD = 0.4). Third, the signal was cut into epochs with the same time range as reported for the EEG signal (from −200 ms to 700 ms around stimulus) and epochs were baseline-corrected (reference window −110 ms to −10 ms before stimulus onset). In the hand-mixed condition, 93.7% of trials remained in the data set on average (range across participants: 1,210 trials to 2,000 trials) and in the foot-mixed condition, 93.6% trials remained (range: 1,193 trials to 1,997 trials).

For the investigation of late potentials, the signals were preprocessed in the same way as described above, except that the reference was kept at the recording reference (at TH6) and the band-pass filter was set to 5 to 400 Hz.

**ENG data preprocessing.**   The peripheral NAPs of interest have very short latencies (i.e., occur almost immediately after the electrical stimulation), meaning that in some participants, the interpolation windows defined at the cervical or lumbar level might be too wide and thus contain the NAPs of interest. Therefore, in order to remove the stimulation artifact, but retain the NAPs, the ENG data were interpolated in a time window from 1.5 ms before to 4 ms after stimulus onset. Data were then down-sampled to 1,000 Hz, band-pass and notch filtered in the same range as ESG data and cut into epochs and baseline-corrected (with the same epoch and baseline windows used for ESG data).

**CCA.**   In order to enhance the SNR and also allow for single-trial analysis, we made use of our multichannel setup and applied CCA to EEG and to the ventral referenced ESG data, separately for the mixed median and tibial nerve stimulation conditions. In the context of EEG, CCA has, for example, been used as blind source separation approach to remove noise such as muscle activity [118] and as a technique to improve single-trial classification of evoked potentials [119]. In both cases, the goal is to obtain a spatial filter and, consequently, a projected component with the largest similarity between two data matrices. Inverting a spatial filter creates corresponding topographies that can then be interpreted in a neurophysiologically meaningful manner [120]. We employed a variant of CCA as used previously for single-trial extraction in EEG data [40–42], also known as canonical correlation average regression [41]. For two multichannel signals $X$ and $Y$, CCA finds the spatial filters $w_x$ and $w_y$ that maximize the correlation

$$\max_{w_x, w_y} \; corr(w_x^T X, w_y^T Y).$$

While both multichannel matrices $X$ and $Y$ have the same size with the structure channel × time, $X$ is a multichannel signal that contains all concatenated epochs from 1 to $N$, and $Y$ is a signal that contains $N$ times the average over all epochs concatenated (with $N$ being the number of all epochs from one participant's recording); in other words, $Y$ is the same size as $X$, only that instead of single trials (as in the case of $X$), it is made up of repetitions of the average of all trials, again using the same latency range as in $X$. More precisely, both $X$ and $Y$ are of size [number of channels × number of samples] and both $w_x$ and $w_y$ are of size [number of channels × number of channels] (in the case of full rank), with "number of channels" being 64 for EEG and 17 for ESG and "number of samples" being N (2,000 in case of no trial rejection) * 11 (see below for rationale). Applied in this way, the CCA procedure serves as a template matching between the single-trial and the average of all trials. The spatial filter $w_x$ corresponds to a spatial weighting of the multichannel signal to separate SEP-related activity from background noise [42]. Since we were interested in early components of the SEP, we only subjected a short time window to CCA (and not the whole epoch length), namely, a window from 5 ms before to 5 ms after the peak of the cortical or spinal SEP component of interest (resulting in 11 data

points per trial). The extracted spatial filter was then applied to the whole length of the epochs. To compute the spatial activity pattern of each CCA component, the spatial filters $w_x$ were multiplied by the covariance matrix of $X$ in order to take the data's noise structure into account [120]. For each stimulation (median or tibial nerve stimulation), one CCA component was selected for further analyses. These components differed in the different data sets and in the different stimulation conditions: In EEG data of median nerve stimulation, the spatial pattern of the selected CCA component corresponded to the typical N20-P35 tangential dipole over the central sulcus and in EEG data of tibial nerve stimulation, it corresponded to the typical P40 radial dipole over medial somatosensory areas. In ESG data of median nerve stimulation, the spatial pattern of the selected CCA component corresponded to a radial dipole (ventral-dorsal direction) over cervical areas as typical for N13, and in ESG data of tibial nerve stimulation, it corresponded to a radial dipole over lumbar areas of the spinal cord as typical for the N22. As expected, the selected component was present in all participants among the first two CCA components, i.e., those with the largest canonical correlation coefficients: For spinal data, we selected the first component in every participant (median first component: $N = 36$; tibial first component: $N = 36$), and for cortical data, we nearly always selected the first component (median first component: $N = 32$; median second component: $N = 4$; tibial first component: $N = 35$; tibial second component: $N = 1$). Because CCA is not sensitive to the polarity of the signal, the spatial filters were multiplied by −1 if necessary, so that the extracted SEP component of interest would always result in the expected peak direction (negative for the cortical N20 and the spinal N13 in the mixed-hand condition, positive for the cortical P40 and negative for the spinal N22 in the mixed-foot condition). Note that for EEG, all channels were subjected to CCA, while for ESG, only channels from the electrode patch of interest were subjected to CCA (i.e., the cervical patch in the hand-mixed condition and the lumbar patch in the foot-mixed condition). Last but not least, it is important to note that for such a multivariate analysis, the number of samples should in principle be at least 10 times the number of variables [121], though more recent efforts also taking into account the effect size suggest an even larger sample-to-feature ratio: e.g., in the case of a between-set correlation of 0.3 (close to the average canonical correlations we observed: 0.25 for median and 0.29 for tibial nerve stimulation) at least 50 samples per feature [122]. In our case, we far exceed the suggested sample-to-feature ratio due to very large number of trials used for training (i.e., in the case of no trial rejections, 2,000 trials with 11 data points each compared to 64 (EEG) or 17 (ESG) channels).

**Brainstem potentials.** Cleaned and epoched EEG and ESG signals, which had been re-referenced during preprocessing to Fz, were combined into one data set and referenced to a common reference at FPz, since frontal channels have been suggested for the investigation of brainstem potentials [27,123,124]. The N14 brainstem potential following median nerve stimulation was extracted from channel SC1 and the N30 brainstem potentials following tibial nerve stimulation was extracted from channel S3 (these potentials have also been described as P14 and P30 in the literature, when using FPz as the active electrode). Please note that we also aimed to apply CCA to brainstem potentials as well but did not succeed.

**Potential amplitude and latency.** For each participant, NAP and SEP latencies were defined individually at the peak of the potential in the average trace over all trials. At the cortical level, SEP latency and amplitude were determined in the CCA component [40–42]. At the spinal level, SEP latency was determined in anatomically defined channels (SC6 for cervical and L1 for lumbar potentials, both thoracic (TH6) referenced) and in the CCA component. Spinal amplitudes were determined in the same channels with thoracic or anterior reference as well as in the cervical or lumbar CCA component. Note that all average traces were visually inspected. In case one of the potentials was not visible in a participant, its latency was estimated based on the average latency of that potential over all participants and the amplitude

was extracted at the estimated latency (Table 1 shows in the column "#" the number of participants in which potentials were detected at the individual level).

**Statistical analysis.** First, to statistically characterize the response in well-known early potentials, we tested peripheral NAP and early SEP peak-amplitudes against zero using one-sample $t$ tests. Second, we investigated whether we might also observe possible later-occurring potentials. For this analysis, we followed the same preprocessing steps, but now filtered with a broader frequency band (5 Hz to 400 Hz), since later components could have lower frequency content. Using resting-state data from the same participants obtained at the very beginning of Experiment 1, we created a surrogate time series with the same stimulation sequence that we preprocessed in the same way. Over a region of interest consisting of the three central columns of the cervical or lumbar electrode grid, we systematically compared the signal from stimulation-runs and from rest-runs in the time window from 0 ms (stimulation onset) to 600 ms using a cluster-based permutation test (in space and time using the FieldTrip toolbox [125]) and focused on responses occurring after the above-reported early potentials (the cluster-based permutation test also identified the N13 and N22, but these are ignored here). In all analyses, significance was established at $p < 0.05$.

**Time-frequency analysis.** For each participant, time-frequency analysis was performed on the averaged trial signal using a continuous short-time fast Fourier transform with a window length of 21 ms and normalized to a baseline interval from 200 ms to 10 ms before stimulus onset. The average over all participants was then displayed.

**SNR.** For all potentials, the SNR was quantified as the root-mean-square of the signal (extracted in a in a time window of +/−1 ms around the individual peak latency) divided by the root-mean-square of the noise (extracted in the same time window before the stimulus onset).

**Assessing the robustness of spinal SEPs.** In order to aid in the planning of future experiments, we assessed the robustness of spinal SEPs as a function of trial number and sample size. Toward this end, we extracted single-trial SEP amplitudes from each participant at the peak latency identified in the average over all trials of that participant, both from anatomically defined channels (with reference at TH6) and from CCA components (trained on the entire data).

Based on these data, we carried out two analyses. First, we assessed the minimum number of trials to obtain a significant result at the level of a *single participant*. For each participant, a subset of trials (trial number varying between 5 and 1,000 in steps of 10, including 1,000) was sampled with replacement, and the significance of amplitudes in the sampled trials was determined using a one-sample $t$ test ($p < 0.05$). This procedure was repeated 1,000 times for each participant, and we report the proportion of significant results for each participant. Second, we determined the minimum number of trials and participants to obtain a significant *group-level* effect. Therefore, we employed Monte Carlo analyses and simulated a large number of experiments[126]. For each "experiment," first, a subset of participants (number varying between 5, 10, 15, 20, 25, 30, 35, 36) was sampled with replacement, and then a subset of trials (number varying between 5 to 1,000 in steps of 10, including 1,000) was sampled with replacement. The trials were then averaged, and a one-sample $t$ test was used to determine the significance. Each experiment was repeated 1,000 times, and we report the proportion of experiments that yielded a significant result (at $p < 0.05$). It is important to note that CCA was only trained once on all trials of mixed-nerve data and then spatial filters were applied to the relevant data, as re-running CCA for each "experiment" was not feasible computationally.

## Data processing and statistical analysis (Experiment 2)

Data processing and analyses followed what is described above for Experiment 1, except that in addition to the hand-mixed and foot-mixed conditions, there were also the hand-sensory (finger1, finger2, fingers1&2) and foot-sensory (toe1, toe2, toes1&2) conditions.

**Stimulation artifact removal.** Identical to Experiment 1, we defined individual artifact windows in cervical and lumbar ESG channels. At the cervical level, average artifact windows ranged from −2.0 ms (std = 1.1 ms) to 4.2 ms (std = 1.8 ms) and at the lumbar level from −2.0 ms (std = 1.1 ms) to 4.8 ms (std = 2.0 ms).

**EEG data preprocessing.** EEG preprocessing was performed in the same way as described above for Experiment 1. One noisy channel was identified in each of six participants and inter-polated before ICA. One difference to the EEG analysis described in Experiment 1 was that in step three, the ICs identified as representing artifactual sources were removed from the EEG signal that (i) consisted of concatenated blocks of each stimulation condition only (i.e., hand-mixed, foot-mixed, hand sensory, or foot-sensory) and (ii) had zero-phase IIR filtering applied with a 50-Hz comb filter (40th order, bandwidth 0.003) and a band-pass (30 to 400 Hz) Butter-worth filter (fourth order); the change in filtering was due to additional line noise and its harmonics introduced by electrical stimulation via ring electrodes. Identical to Experiment 1, noisy time points were removed, but here, this did not result in the exclusion of additional channels. In Experiment 2, epochs were cut from 200 ms before to 300 ms after stimulus onset and baseline-corrected (with a reference interval from −110 ms to −10 ms before stimulus onset). Across conditions, this procedure resulted in the following number of trials remaining on average: hand-sensory 99.5% (range across participants: 5,795 trials to 6,000 trials), hand-mixed 99.4% (range across participants: 1,921 trials to 2,000 trials), foot-sensory 99.2% (range across participants: 5,678 trials to 6,000 trials), and foot-mixed 99.8% (range across partici-pants: 1,978 trials to 2,000 trials).

**ESG data preprocessing.** Since ESG data were preprocessed the same way as described in Experiment 1, only the differences are listed in the following. After cardiac artifact correction, an average of 1.8 channels (std = 1.0) were removed in four participants. Due to the use of ring electrodes for digit stimulation, more line noise and its harmonics were visible in the data. Therefore, zero-phase IIR filtering was applied with a 50-Hz comb filter (40th order, band-width 0.003) and a band-pass (30 to 400 Hz) Butterworth filter (fourth order). Similar to Experiment 1, time points with ESG activity above 100 μV were removed from the continuous data, and if more than 50% of data points were removed from a channel, the whole channel was excluded instead. In one participant, two additional channels were removed. The signal was cut into epochs with the same time range as reported for the EEG signal (from −200 ms to 300 ms around stimulus onset), and epochs were baseline-corrected (reference window −110 ms to −10 ms before stimulus onset). On average, 91.3% of trials remained in the hand-mixed condition (range across participants: 999 trials to 2,000 trials), 90.5% of trials remained in the hand-sensory conditions (range across participants: 3,873 trials to 5,993 trials), 94.2% of trials remained in the foot-mixed condition (range across participants: 1,433 trials to 2,000 trials), and 91.4% of trials remained in the foot-sensory conditions (range across participants: 3,751 trials to 5,988 trials).

**ENG data preprocessing.** ENG data were processed the same way as described for Experi-ment 1 above.

**CCA.** CCA was trained in the same way as explained above for Experiment 1. More spe-cifically, it was trained on data from mixed nerve conditions (due to their higher SNR), and the spatial filters were then applied to the respective mixed and sensory nerve conditions. The selected component was present in all participants among the first 2 CCA components, i.e.,

those with the largest canonical correlation coefficients: For spinal data, we selected the first component in every participant (median first component: $N = 24$; tibial first component: $N = 24$), and for cortical data, we nearly always selected the first component (median first component: $N = 20$; median second component: $N = 4$; tibial first component: $N = 22$; tibial second component: $N = 2$).

**Brainstem potentials.** We did not investigate brainstem potentials in Experiment 2 due to the lower SNR of SEPs after sensory nerve stimulation.

**Potential amplitude and latency.** These metrics were calculated in identical fashion as described for Experiment 1.

**Statistical analysis.** SEP amplitudes from all experimental conditions were compared against zero using one-sample *t* tests. SEP amplitudes and latencies in mixed and sensory conditions were compared using paired *t* tests. To balance the number of stimuli for mixed and sensory conditions, only the double stimulation conditions were subjected to this statistical comparison.

**SNR.** For all potentials, the SNR was quantified as the root-mean-square of the signal (extracted in a in a time window of +/−1 ms around the individual peak latency) divided by the root-mean-square of the noise (extracted in the same time window before the stimulus onset).

**Assessing the robustness of spinal SEPs.** In order to also assess the robustness of the spinal SEPs elicited by sensory nerve stimulation, we repeated the same analyses as outlined for Experiment 1, though this time for the conditions finger1, finger2, fingers1&2, toe1, toe2, and toes1&2. Please note that we adjusted the number of participants (number varying between 5, 10, 15, 20, 24) according to the smaller sample size of Experiment 2.

**Linear-mixed-effects models across somatosensory processing levels.** To examine whether electrophysiological signals covaried across different stages of somatosensory processing, we employed LME models. Specifically, we tested whether the effect of stimulation condition (mixed nerve, finger/toe1, finger/toe2, fingers/toes1&2) on signal amplitude propagated through the somatosensory processing hierarchy. For this, we used random-intercept LME models with the random factor participant, and in- or excluding the factor stimulation condition (with mixed nerve as reference level) to the regressions of peak amplitudes on consecutive somatosensory processing levels in the following way:

$$spinal\ cord \sim 1 + periphery + (1|participant)$$

$$spinal\ cord \sim 1 + periphery * condition + (1|participant)$$

$$S1 \sim 1 + spinal\ cord + (1|participant)$$

$$S1 \sim 1 + spinal\ cord * condition + (1|participant).$$

These analyses were separately performed for stimulation conditions of the hand and the foot. Variables "spinal cord" and "S1" correspond to the single-trial peak amplitudes of the respective signals extracted using CCA as explained in the methods section "CCA," and "periphery" to the peripheral single-trial NAP peak amplitude measured at the axilla or popliteal fossa in hand and foot stimulation, respectively (in foot stimulation, the signal was derived from the knee electrode with the largest evoked potential). All amplitude measures were z-transformed before including them in the LME models. The fixed-effect coefficients were estimated based on the maximum likelihood (ML), and *p*-values of the fixed-effect coefficients were obtained adjusting the denominator degrees of freedom according to Satterthwaite's

method [127]. The LME models were calculated in R (version 4.2.0 [128]) with the lmer function of the lme4 package (version 1.1–30 [129]), as well as including the lmerTest package (version 3.1–3 [130]) for the implementation of the Satterthwaite method.

**IR.** If the information from the simultaneous stimulation of 2 digits (fingers or toes) is integrated at a certain neural processing stage, then the SEP amplitude following this simultaneous digit stimulation should be reduced compared to arithmetic sum of the SEP amplitudes following separate stimulation of the two digits. To quantify this attenuation effect for each participant, we calculated an IR as suggested previously [44,45,131]. The IR captures the amplitude attenuation caused by the simultaneous stimulation of two digits and describes this attenuation as percentage of the expected amplitude sum of single-digit stimulations:

$$IR = \left( \sum (D1, D2) - D1D2 \right) / \sum (D1, D2) * 100$$

where $\Sigma(D1,D2)$ is the sum over SEP (or NAP) amplitudes following single-digit (finger/toe1 or finger/toe2) stimulation and D1D2 the SEP (or NAP) amplitude following double-digit stimulation (fingers/toes1&2). A positive IR would reflect the percentage of SEP amplitude attenuation from the expected amplitude (i.e., the sum of SEP amplitudes to single-digit stimulation), and an IR of 0% would suggest that there is no integration happening, meaning SEP amplitudes to double-digit and the sum of single-digit stimulations have the same size (a negative IR would mean that there is an amplification effect of SEP amplitudes to double-digit stimulation). IR values from each participant to finger and toe stimulation were tested against zero using one-sample *t* tests.

### Experiment 3: Nociceptive stimulation

**Participants.** We acquired data from seven healthy volunteers (five female; mean age: 30.6 years, range: 23 to 36 years), all of whom provided written informed consent. The study was approved by the Ethics Committee at the Medical Faculty of the University of Leipzig.

**Laser stimulation.** Individually calibrated painful heat stimuli (duration 125 ms) were delivered to the dorsum of left hand using a $CO_2$-laser with a wavelength of 10.6 μm and a beam diameter of 6 mm (LSD; Laser Stimulation Device, SIFEC s.a., Ferrières, Belgium). The LSD contains a closed loop temperature control system to maintain constant skin temperature during stimulation by adjusting the energy output. The stimulus position was controlled by an electric motor moving the laser head relative to a participant's hand, allowing for precise control of stimulation position. Throughout the entire experiment, participants wore protective goggles.

**Experimental design.** The here-reported data are part of a larger experiment also involving other stimulation modalities, but we solely focus on laser stimulation in this report. Before any electrodes were attached to the participant, the experiment started with a calibration procedure in order to find temperatures that would be perceived as clearly painful but tolerable (mean temperature: 55.9°C; range: 53 to 59°C). There were 10 blocks of laser stimulation (with a break of approximately 5 to 10 minutes between blocks), with each block containing 36 stimuli, separated by an ISI of 1.53 seconds with a jitter between +/−100 ms (drawn from a uniform distribution). In each block, the laser beam was shifted over the dorsum of the left hand in an S-shaped pattern along a $6 \times 6$ grid (size $5 \times 5$ cm): The start could be in any of the four corners of the grid, and the laser would always move along the rows in the anterior/posterior direction before moving to the next column, until all 36 cells had been stimulated once.

**Electrographic recordings.** ECG, EEG, EOG, and ESG data were acquired using the same equipment as described in Experiment 1 and Experiment 2. ECG data were recorded via an electrode placed on the left costal arch, referenced to an electrode placed underneath the right

clavicular. EEG data were recorded via a standard 32-channel montage according to the 10–20 system and referenced to the nose. EOG data were recorded via two additional electrodes placed on the canthus of the right eye (referenced to nose) and below the right eye (referenced to Fp2). ESG recordings were again based on a custom-made electrode patch (consisting of the same fabric as the EEG cap), but now with a higher electrode number than in Experiment 1 and Experiment 2 and focused solely on the cervical spinal cord. The patch consisted of 38 electrodes centered around an electrode over the spinous process of the seventh cervical vertebra. The midline of this electrode-grid was positioned vertically along the spine and consisted of seven electrodes (the fourth one being centered on vertebra C7) with a vertical interelectrode distance of 2 cm. Two further vertical lines of six electrodes each were placed 1.5 cm to the right and left of the midline electrodes, another two vertical lines of five electrodes were placed 3 cm to the right and left of the midline, and another two vertical lines of two electrodes each were placed 5 cm to the right and left of the midline. Additional electrodes were placed on the first cervical vertebra and on the inion. In addition to the dorsal electrodes, there were also three ventral electrodes at the anterior neck (one supraglottic electrode (CA1), one above the suprasternal notch (CA3), and the third one in the middle between these two (CA2)). ESG data were referenced to an electrode positioned over the spinous process of the sixth thoracic vertebra (Th6). The active ground electrode stabilized the signal via the "driven right leg" principle. It was placed at POz in the EEG montage and on the spinous process of the 10th thoracic vertebra in the ESG montage.

**Data analysis—EEG.** All analyses were performed using Python 3.10 and MNE (https://mne.tools/stable/index.html; version 1.6.0). Data from the 10 experimental blocks were concatenated and down-sampled to 500 Hz. Down-sampled data were then high-pass filtered at 1 Hz (using a fourth order Butterworth filter, effective order 8) and notch filtered around 50 Hz and harmonics (using an eighth order Butterworth filter, effective order 16). Subsequently, data were epoched in a window between 300 ms before and 1,000 ms after stimulus onset. Invalid trials (i.e., aborts of the laser) were removed, followed by a manual removal of extremely noisy epochs as determined by visual inspection. Data were further low-pass filtered with a cutoff frequency of 30 Hz (fourth order Butterworth filter) and either re-referenced to the average of all EEG electrodes (for analysis of the N2P2 complex) or to Fz (for analysis of the N1).

**Data analysis—ESG.** All analyses were performed using Python 3.10 and MNE (https://mne.tools/stable/index.html; version 1.6.0). First, in order to remove possible artifacts resulting from stimulation, data were linearly interpolated between −13 ms and 13 ms relative to stimulus onset. Data were then down-sampled to 1 kHz and notch filtered to remove powerline noise at 50 Hz and all harmonics up to 200 Hz with an IIR filter. Next, the cardiac artifact was removed using signal space projection with six projectors, and the data were band-pass filtered from 30 Hz to 150 Hz using a fourth order Butterworth, zero-phase filter. The data were then epoched from −100 ms to 300 ms relative to stimulation, with the baseline period defined from −100ms to −10 ms. Finally, CCA was applied as described previously for Experiment 1 and Experiment 2, with the onset and duration of the training window changed based on the following reasoning.

Surface recordings show that with appropriate task/analysis, one can observe laser-evoked cortical responses peaking as soon as 83 ms (EEG data; [132]) or 98 ms (MEG-data; onset at 84 ms; [133]) after stimulation onset, with invasive recordings revealing that the onset of cortical responses to laser stimulation can be early as approximately 70 ms in S1 [134]. Invasive thalamic recordings demonstrate spikes between 60 and 70 ms [135] and induced responses in the gamma range at approximately 90 ms [136] after laser stimulation (but see [137] for later responses). fMRI-EEG fusion results points toward thalamic responses to laser stimulation

from 65 ms (in a lateral nucleus) and 89 ms (in a medial nucleus) onwards [138]. Together, these evoked-response data suggest that initial spinal responses could occur even before 60 ms. With respect to conduction-velocity data, estimates of human spinothalamic tract conduction velocity vary between laboratories and employed methods (see [139] and responses thereto) and have been shown to differ in various spinothalamic pathways [140–142]. We thus did not base our estimation as to when to expect spinal responses on these estimates but instead additionally relied on peripheral nerve conduction velocity estimates of A-delta fibers mediating responses to laser stimulation. These have been estimated to vary between 9 and 18 m/s ([143]: 9 m/s; [144]: 11 m/s; [145]: 13 m/s; [146]: 16 m/s; [135]: 18 m/s) and thus suggest possible initial spinal responses to occur roughly 45 ms to 90 ms after stimulation when (i) considering an approximate distance of 80 cm between hand dorsum and spinal cord and (ii) ignoring any delay between laser stimulation onset and action potential generation in the peripheral nerve. Based on the above considerations, we trained CCA on a time-window of 45 ms to 90 ms after laser stimulation onset—this is only a heuristic for this first proof-of-principle experiment, and it is likely that future studies investigating electrophysiological spinal responses in much more detail might lead to more optimized training windows.

For each participant, the first CCA component (as ranked by their canonical correlation coefficient) was selected, and the resulting time-courses were averaged across participants to obtain a group-average response. Since CCA is not sensitive to the polarity of the signal and since we observed a negative deflection at approximately 50 ms in the component time-course in six out of seven participants (but a positive deflection at this time-point in participant 2), we multiplied this participant's time-course by -1 and used this sign-inverted time-course in all further analyses (similar to the procedure used in Experiment 1 and Experiment 2). In order to demonstrate the robustness of the obtained results, we also performed a within-participant four-fold split of the data (i.e., first split: trials 1,5,9,. . .; second split: trials 2,6,10,. . .; third split: trials 3,7,11,. . .; fourth split: trials 4,8,12,. . .) after having applied the spatial filter and then averaged the results of each fold across participants.

## Open science

Experiment 1 and Experiment 2 were preregistered on the Open Science Framework before the start of data acquisition and the preregistrations are openly available (see https://osf.io/sgptz and https://osf.io/mjdha); differences between the analyses suggested in the preregistrations and the analyses carried out here are listed in S1 Text. All data are openly available (https://openneuro.org/datasets/ds004388, https://openneuro.org/datasets/ds004389, https://openneuro.org/datasets/ds005307) in EEG-BIDS format [147,148]. All analysis code has been deposited on GitHub and is openly available (see https://github.com/eippertlab/spinal_sep1, https://doi.org/10.5281/zenodo.13383050; https://github.com/eippertlab/spinal_sep2, https://doi.org/10.5281/zenodo.13383046; https://github.com/eippertlab/spinal_lep1, https://doi.org/10.5281/zenodo.13383056).

## Supporting information

**S1 Text.** This file contains descriptions regarding (i) analysis differences between preregistration and manuscript; (ii) results for mixed nerve stimulation from Experiment 2; (iii) results for the later spinal SEP components from Experiment 2; (iv) results from the sensory nerve stimulation in Experiment 2; (v) results from the analysis on changes in response amplitude across the processing hierarchy; and (vi) the choice of reference electrode.
(PDF)

**S1 Fig.** Grand-average over all participants in the foot-mixed condition and in simulated epochs from rest data. The plotted signal is an average over all channels that are part of the identified cluster (channels displayed as red dots on the top left). The gray area between 126–132 ms identifies the time range in which the 2 signals are statistically different; note that this result did not replicate in Experiment 2.
(TIF)

**S1 Table.** Group-level descriptive statistics for SEP- and NAP-amplitudes, latencies and SNR (mean and standard error of the mean) and one-sample t test of SEP- and NAP-amplitudes in the hand-mixed and foot-mixed conditions of Experiment 2 (N = 24). Note that we only focused on the major peripheral, spinal, and cortical components here for replication purposes and thus do not report Erb's point and brainstem potentials. Abbreviations: vr = ventral reference, tr = thoracic reference, CCA–canonical correlation analysis, SEP = somatosensory evoked potential, NAP = nerve action potential, # = number of participants in which potential was visible at the individual level, SNR = signal-to-noise ratio).
(PDF)

**S2 Table.** Group-level descriptive statistics for SEP- and NAP-amplitudes, latencies and SNR (mean and standard error) and one-sample t test of SEP- and NAP-amplitudes in all hand-sensory and foot-sensory conditions of Experiment 2 (vr = ventral reference, tr = thoracic reference, CCA = canonical correlation analysis, # = number of participants with potentials visible at the individual level).
(PDF)

**S3 Table.** Paired t test for the comparisons between hand-mixed and fingers1&2 conditions or foot-mixed and toes1&2 conditions. Tested were the amplitudes and the latencies of the SEPs and peripheral NAPs. Data come only from Experiment 2 (vr = ventral reference, tr = thoracic reference, CCA = canonical correlation analysis).
(PDF)

**S1 Data.** Each sheet (5 sheets for Figure 2A and 5 sheets for Figure 2B) contains every participant's electrophysiological data (ENG, ESG, EEG), which together give rise to 1 grand-average trace reported in the figure. Each row reflects 1 time point, and each column represents the amplitude of 1 participant's average evoked response.
(XLSX)

**S2 Data.** Sheets Figure_3A and Figure_3E. Each sheet contains every participant's ESG data, which together give rise to 1 grand-average trace reported in the figure. Each row reflects 1 time point, and each column represents the amplitude of 1 participant's average evoked response, first for single-channel data and then for CCA data. Sheets Figure_3B and Figure_3F. Each sheet contains every participant's ESG data, which together give rise to the grad-average isopotential plots. Each row corresponds to 1 channel, and each column corresponds to 1 participant. Sheets Figure_3C_sub-1-18, Figure_3C_sub-19-36, Sheets Figure_3G_sub-1-18, Figure_3G_sub-19-36. Each sheets contains every participant's ESG data, which together give rise to 1 grand-average time-frequency plot. The rows correspond to frequencies, and the columns correspond to time points per participant. Please note that we had to create 2 sheets for each panel due to data-size limitations. Sheets Figure_3D and Figure_3H. Each sheet contains every participant's ESG data, which together give rise to 1 grand-average trace reported in the figure. Each row reflects 1 time point, and each column represents the amplitude of 1 participant's average evoked response (average over all channels that are part of the identified cluster), first for stimulation and then for resting-state data. The last column depicts time points of

significant difference as established by the cluster-based permutation test.
(XLSX)

**S3 Data.** Sheets Figure_4A and Figure_4E. Each sheet contains the ESG-SNR values of each participant (columns), with single-channel data being represented in the upper row and CCA data being represented in the lower row. Remaining 6 sheets. Each sheet contains the individual participant ESG data underlying panels B-D and F-H (sub-006, sub-014, sub-021), with the first 3 sheets depicting cervical data and the last 3 sheets depicting lumbar data. Each row corresponds to 1 trial, and each column corresponds to 1 time point, first presented for the single-channel data and then for the CCA data.
(XLSX)

**S4 Data.** Each sheet (1 sheet for Figure 5A and 1 sheet for Figure 5B) contains every participant's ESG data, which together give rise to 1 grand-average trace reported in the figure. Each row reflects 1 time point, and each column represents the amplitude of 1 participant's average evoked response after CCA, first for mixed and then for sensory nerve data.
(XLSX)

**S5 Data.** Each sheet (3 sheets for Figure 6A and 3 sheets for Figure 6B) contains every participant's electrophysiological data (ENG, ESG, EEG), which together give rise to 1 grand-average trace reported in the figure. Each row reflects 1 time point, and each column represents the amplitude of 1 participant's average evoked response after CCA, first for digit 1 stimulation, then for digit 2 stimulation and finally for digit 1&2 stimulation.
(XLSX)

**S6 Data.** Sheets Figure_8A_Upper and Figure_8A_Lower. Depicted are EEG data, with each row reflecting 1 time point and each column representing the amplitude of 1 participant's average evoked response from 1 electrode. Sheet Figure_8B. Depicted are ESG data, with each row reflecting 1 time point and each column representing the amplitude of 1 participant's average evoked response of component 1 after CCA for each given time point. Sheet Figure_8C. Depicted are ESG data, with each row reflecting 1 time point and each column representing the amplitude of 1 participant's evoked response based on the following subsets of trials in component 1 after CCA: _1: every fourth available trial beginning with trial 1; _2: every fourth available trial beginning with trial 2; _3: every fourth available trial beginning with trial 3; _4: every fourth available trial beginning with trial 4. Sheet Figure_8D. Depicted are ESG data, with the data being the same as those displayed in panel B, though in this case rather than plotting the average across all participants, each panel depicts 1 participant's evoked response across all trials.
(XLSX)

**S7 Data.** This sheet contains every participant's ESG data, which together give rise to 1 grand-average trace reported in the figure. Each row reflects 1 time point, and each column represents the amplitude of 1 participant's average evoked response (average over all channels that are part of the identified cluster), first for stimulation and then for resting-state data. The last column depicts time points of significant difference as established by the cluster-based permutation test.
(XLSX)

## Acknowledgments

We would like to thank our student research assistants Janek Haschke, Pia-Lena Baisch, Paula Kosel, Max Braune, Samuel Simeon, and Marleen Löffler for their help in recruitment and data acquisition.

## Author Contributions

**Conceptualization:** Birgit Nierula, Vadim V. Nikulin, Falk Eippert.

**Data curation:** Birgit Nierula.

**Formal analysis:** Birgit Nierula, Tilman Stephani, Emma Bailey, Merve Kaptan.

**Funding acquisition:** Falk Eippert.

**Investigation:** Birgit Nierula, Lisa-Marie Geertje Pohle.

**Methodology:** Birgit Nierula, Emma Bailey, Lisa-Marie Geertje Pohle, Ulrike Horn, André Mouraux, Gabriel Curio, Vadim V. Nikulin, Falk Eippert.

**Project administration:** Birgit Nierula, Falk Eippert.

**Resources:** Arno Villringer, Vadim V. Nikulin, Falk Eippert.

**Software:** Birgit Nierula, Tilman Stephani, Emma Bailey, Merve Kaptan.

**Supervision:** Vadim V. Nikulin, Falk Eippert.

**Visualization:** Birgit Nierula, Falk Eippert.

**Writing – original draft:** Birgit Nierula, Falk Eippert.

**Writing – review & editing:** Birgit Nierula, Tilman Stephani, Emma Bailey, Merve Kaptan, Lisa-Marie Geertje Pohle, Ulrike Horn, André Mouraux, Burkhard Maess, Arno Villringer, Gabriel Curio, Vadim V. Nikulin, Falk Eippert.

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
