## [Editor Report · Decision Letter 0]

15 Feb 2024

Dear Dr Eippert, 

Thank you for submitting your manuscript entitled "Closing the gap between brain and body: non-invasive multi-channel electrophysiology enables human spinal cord imaging with single-trial precision" for consideration as a Research Article by PLOS Biology. Please accept my apologies for the delay in getting back to you as we consulted with an academic editor about your submission. 

Your manuscript has now been evaluated by the PLOS Biology editorial staff, as well as by an academic editor with relevant expertise, and I am writing to let you know that we would like to send your submission out for external peer review.

IMPORTANT - After discussions within the editorial team, we think that your manuscript would be a better fit as a Methods and Resources Article. Upon resubmission (see below), we would be grateful if you could please tick 'Methods and Resources' as the article type in the dropdown menu. 

Before we can send your manuscript to reviewers, we need you to complete your submission by providing the metadata that is required for full assessment. To this end, please login to Editorial Manager where you will find the paper in the 'Submissions Needing Revisions' folder on your homepage. Please click 'Revise Submission' from the Action Links and complete all additional questions in the submission questionnaire.

Once your full submission is complete, your paper will undergo a series of checks in preparation for peer review. After your manuscript has passed the checks it will be sent out for review. To provide the metadata for your submission, please Login to Editorial Manager (https://www.editorialmanager.com/pbiology) within two working days, i.e. by Feb 17 2024 11:59PM.

Kind regards,

Richard

Richard Hodge, PhD

rhodge@plos.org

PLOS

---

## [Decision Letter · Decision Letter 1]

24 Mar 2024

Dear Dr Eippert,

Thank you for your patience while your manuscript "Closing the gap between brain and body: non-invasive multi-channel electrophysiology enables human spinal cord imaging with single-trial precision" was peer-reviewed at PLOS Biology as a Methods and Resources Article. Please accept my sincere apologies for the delays that you have experienced during the peer review process. Your manuscript has now been evaluated by the PLOS Biology editors, an Academic Editor with relevant expertise, and by two independent reviewers. 

In light of the reviews, which you will find at the end of this email, we would like to invite you to revise the work to thoroughly address the reviewers' reports.

As you will see, the reviewers are generally very positive about the approach and think it is well-designed and an important contribution. However, Reviewer #1 asks that additional reporting details are provided and notes that the limitations around the user selection of the time period should be discussed. In addition, Reviewer #2 asks that additional proof-of-concept studies are provided to fully demonstrate the utility of the approach. After discussions with the Academic Editor, we also ask that you please provide a comparison with other recent papers based on fMRI recordings in the discussion section. 

Given the extent of revision needed, we cannot make a decision about publication until we have seen the revised manuscript and your response to the reviewers' comments. Your revised manuscript is likely to be sent for further evaluation by all or a subset of the reviewers.

**IMPORTANT - SUBMITTING YOUR REVISION**

*Re-submission Checklist*

*Published Peer Review*

*PLOS Data Policy*

*Blot and Gel Data Policy*

Sincerely,

Richard

Richard Hodge, PhD

rhodge@plos.org

REVIEWS:

Reviewer #1: This is an important paper which outlines electrophysiological recordings along the complete neuro-axis from peripheral nerve , through brainstem and to primary sensor cortex. 

It highlights the quality of work that can be done with standard and affordable electrophysiological measures.

This represents a huge amount of recording time and number of subjects in a well designed paradigm.

I do have one major concern, that is hopefully a misunderstanding, and that concerns the CCA analysis.

Firstly, as the authors point out, there is the user selection of the time-period and then a choice of the significant CCA component. This adds to some loss of objectivity. This should be noted in the limitations section. As the time-window was user specified, please make it clear how SNR was calculated (i.e. if you choose the peak to optimize, and then measure SNR at that peak is there some circularity).

Please make it clear how CCA was used in the Interaction ration section. For example, it would seem that amplitudes can only be compared if the same linear (CCA) mixture is used for all comparisons (rather than a different CCA mixture for each component).

Secondly, could the authors please be more explicit with the notation on page 22. 

Please give the dimensions of X and Y and the wx, wy in terms of rows and columns. 

Also please specify Y in terms of X . As far as I understand Y is a repetition of a specific latency range of mean(X) over Ntrials.

Then also please specify (explicitly in equation) how the canonical component of interest is created

My confusion is that normally one would have more (at least 4 times) trials (rows) than features (columns).

My understanding is that, by definition, one would expect a very high and significant canonical correlations otherwise.

Again, hopefully adding more information (like effective degrees of freedom and as many other details of the CCA as possible etc) should remove these concerns.

One way to show this is less of a worry empirically could be to replace the raw data X with the resting state data (as used elegantly to demonstrate the late component in figure 3G). Ideally this would not show a significant component. 

Minor.

Figure 1. Please put number of subjects that went into grand average into figure caption.

Reviewer #2: The paper shows the optimization of a methodology to record spinal cord activities in a non invasive way. The demonstration of the efficacy of the approach is well-designed and the results quite convincing. However, I found a bit limited the overall investigation of the real usability of this approach. I would suggest the authors to add some experiments with more "ecological" touch or proprioceptive stimuli of hand and foot...what is the overall ability to discriminate these sensations? it is able to provide results which are anatomically plausible etc. This kind of analysis would make more convincing the overall translational possibility of this approach.

---

## [Decision Letter · Decision Letter 2]

14 Aug 2024

Dear Dr Eippert,

Thank you for your patience while we considered your revised manuscript "Closing the gap between brain and body: non-invasive multi-channel electrophysiology enables human spinal cord imaging with single-trial precision" for publication as a Methods and Resources Article at PLOS Biology. This revised version of your manuscript has been evaluated by the PLOS Biology editors, the Academic Editor and the original reviewers.

Based on the reviews, I am pleased to say that we are likely to accept this manuscript for publication, provided you address the following data and other policy-related requests that I have provided below (A-F):

(A)We would like to suggest the following minor modification to the title:

“A multi-channel electrophysiology approach to non-invasively and precisely record human spinal cord activity”

(B) You may be aware of the PLOS Data Policy, which requires that all data be made available without restriction: http://journals.plos.org/plosbiology/s/data-availability. For more information, please also see this editorial: http://dx.doi.org/10.1371/journal.pbio.1001797

-Supplementary files (e.g., excel). Please ensure that all data files are uploaded as 'Supporting Information' and are invariably referred to (in the manuscript, figure legends, and the Description field when uploading your files) using the following format verbatim: S1 Data, S2 Data, etc. Multiple panels of a single or even several figures can be included as multiple sheets in one excel file that is saved using exactly the following convention: S1_Data.xlsx (using an underscore).

-Deposition in a publicly available repository. Please also provide the accession code or a reviewer link so that we may view your data before publication. 

Figure 2A-B, 3A-H, 4A-H, 5A-B, 6A-B, 7A-H, 8A-D, S1

(C) Please make the data uploaded in EEG-BIDS format to OpenNeuro publicly available at this stage. 

(D) Please also ensure that each of the relevant figure legends in your manuscript include information on *WHERE THE UNDERLYING DATA CAN BE FOUND*, and ensure your supplemental data file/s has a legend.

(E) Please ensure that your Data Statement in the submission system accurately describes where your data can be found and is in final format, as it will be published as written there. 

(F) Please note that we cannot accept sole deposition of code in GitHub, as this could be changed after publication. However, you can archive this version of your publicly available GitHub code to Zenodo. Once you do this, it will generate a DOI number, which you will need to provide in the Data Accessibility Statement (you are welcome to also provide the GitHub access information). See the process for doing this here: https://docs.github.com/en/repositories/archiving-a-github-repository/referencing-and-citing-content

We expect to receive your revised manuscript within two weeks. 

*Published Peer Review History*

*Press*

Kind regards,

Richard

Richard Hodge, PhD

rhodge@plos.org

Reviewer remarks:

Reviewer #1: I would like to thank the authors for comprehensively addressing my concerns.

---

## [Editor Report · Decision Letter 3]

2 Sep 2024

Dear Falk,

On behalf of my colleagues and the Academic Editor, Simon Hanslmayr, I am pleased to say that we can accept your manuscript for publication, provided you address any remaining formatting and reporting issues. These will be detailed in an email you should receive within 2-3 business days from our colleagues in the journal operations team; no action is required from you until then. Please note that we will not be able to formally accept your manuscript and schedule it for publication until you have completed any requested changes.

During the production process, I would also be grateful if you could please ensure that the OpenNeuro data depositions are made publicly available. The weblink provided in the Data Availability Statement for Study 3 appears to work fine, but the links for Study 1 and 2 do not? 

PRESS

Best wishes, 

Richard

Richard Hodge, PhD

rhodge@plos.org

PLOS
